# Mild photothermal therapy potentiates anti-PD-L1 treatment for immunologically cold tumors via an all-in-one and all-in-control strategy

Liping Huang[1], Yanan Li[2], Yunai Du[2], Yiyi Zhang[1], Xiuxia Wang[1], Yuan Ding[2], Xiangliang Yang[1,3], Fanling Meng[1,3], Jiasheng Tu[2]*, Liang Luo [1,3]* & Chunmeng Sun [2]*

One of the main challenges for immune checkpoint blockade antibodies lies in malignancies with limited T-cell responses or immunologically "cold" tumors. Inspired by the capability of fever-like heat in inducing an immune-favorable tumor microenvironment, mild photothermal therapy (PTT) is proposed to sensitize tumors to immune checkpoint inhibition and turn "cold" tumors "hot." Here we present a combined all-in-one and all-in-control strategy to realize a local symbiotic mild photothermal-assisted immunotherapy (SMPAI). We load both a near-infrared (NIR) photothermal agent IR820 and a programmed death-ligand 1 antibody (aPD-L1) into a lipid gel depot with a favorable property of thermally reversible gel-to-sol phase transition. Manually controlled NIR irradiation regulates the release of aPD-L1 and, more importantly, increases the recruitment of tumor-infiltrating lymphocytes and boosts T-cell activity against tumors. In vivo antitumor studies on 4T1 and B16F10 models demonstrate that SMPAI is an effective and promising strategy for treating "cold" tumors.

---

[1] National Engineering Research Center for Nanomedicine, College of Life Science and Technology, Huazhong University of Science and Technology, Wuhan 430074, China. [2] Center for Research Development and Evaluation of Pharmaceutical Excipients and Generic Drugs, Department of Pharmaceutics, School of Pharmacy, China Pharmaceutical University, 24 Tong Jia Xiang, Nanjing 210009, China. [3] Hubei Key Laboratory of Bioinorganic Chemistry and Materia Medica, School of Chemistry and Chemical Engineering, Huazhong University of Science and Technology, Wuhan 430074, China. *email: jiashengtu@aliyun.com; liangluo@hust.edu.cn; suncm_cpu@hotmail.com

I mmune checkpoint blockade (ICB) that aims to reverse signals from the immunosuppressive tumor microenvironment (TME) is being driven as a primary treatment modality[1–3]. The efficacy of ICB-based cancer immunotherapy largely depends on the expression of PD-L1 in the tumor tissues and the recruitment of tumor-infiltrating lymphocytes (TILs)[4–9]. However, recent findings showed that many tumors do not contain high levels of TILs seemingly required to receive clinical benefit[10]. These tumors, which are considered as immunologically "cold"[11,12], do not respond to immune checkpoint inhibitors[13–15]. Although enhanced anti-PD therapy efficacy has been achieved in treating multiple tumor types in combination with various therapeutic strategies, such as chemotherapy[16,17], phototherapy[18,19], radiotherapy[9,20], and other immunotherapies[21,22], strategies that can directly elevate the level of TILs in the TME, i.e., convert a "cold" tumor to a "hot" tumor, are still required to increase the potential responses to ICB.

Photothermal therapy (PTT) with the advantages of localized treatment, noninvasiveness, and controllable irradiation and temperature has emerged as a new paradigm toward precise cancer therapy[17,21,22]. Typically, to achieve a relatively harsh environment for efficient ablation of tumors, rigorous photothermal heating to high temperature over 50 °C is required[23], which will also destroy normal tissues[24]. However, once the photothermal energy is reduced, the outcomes of PTT are dramatically impaired and the PTT-induced abscopal effect is too weak to suppress the growth of the remaining tumor margin[25]. Recently, we noticed that mild PTT with a relatively low temperature at ~ 45 °C was applied in tumor treatment as an aid rather than directly killing tumor cells[25–27]. Theoretically, mild heating is considered as a critical variable in producing a favorable TME for immunological responses[28]. However, such a small elevation in temperature can also upregulate some proteins, such as heat shock protein (HSP), indoleamine 2,3-dioxygenase, and PD-L1, on tumor cells for self-protection and turn them immunosuppressive[25,29]. In this situation, it has inspired us to propose that the combination of mild PTT and anti-PD therapy will be a potential strategy to overcome the drawbacks of both mild PTT and immunotherapy, which may increase the immunogenicity of tumors to reprogram the "cold" TME and sensitize these tumors to immune checkpoint inhibition for synergistic anticancer therapy.

To validate the above hypothesis, an injectable lipid gel (LG) depot engineered with the ability of photothermal sensitivity and reversible formation change was adopted as a two-pronged modality for localized symbiotic mild photothermal-sensitized immunotherapy (SMPAI) (Fig. 1). We encapsulate both payloads, i.e., a photothermal agent (IR820) and an anti-PD-L1 antibody (aPD-L1), into a lipid mixture (LG precursor) of soybean phosphatidylcholine (SPC) and glycerol dioleate (GDO) in an all-in-one manner. After this formulation is intratumorally injected, the LG precursor hydrates into a gel depot (aPD-L1/I@LG) that is able to undergo a reversible gel-to-sol transition in response to a small temperature change at around 39 °C. IR820 under manually controlled NIR irradiation can generate mild heat in the injection site and induce a tunable LG phase change for the controllable release of aPD-L1. Meanwhile, the mild PTT can activate the systemic immune response to increase the tumor infiltration of T cells and upregulate the expression of PD-L1 on the surface of tumor cells, which is expected to effectively remodel the TME and sensitize tumors to ICB therapeutic immune response. This all-in-control strategy emphasizes the controllable therapeutic release and long-term antitumor effect, and the desired pre-programmed dosage regimen can be achieved according to the various needs of carcinoma-suffering individuals. Here we apply this combined all-in-one and all-in-control strategy to confirm the improvement

of anti-PD-L1 therapy in two tumor models with scarce TILs, 4T1, and B16F10[8,29,30]. For 4T1 models, we examine the therapeutic effect on both primary and distal tumors. The successful prevention of tumor growth in both tumor models demonstrates that this SMPAI strategy may be generally applicable to improve immune response of immunologically "cold" tumors.

## Results

**Preparation and characterization of in situ thermal-responsive LG.** We employed the Macrosol technique to encapsulate the hydrophilic photosensitizer IR820 and aPD-L1 to the lipid precursor of the LG. A lyophilized mixture of aPD-L1 and IR820 was distributed homogeneously in pre-mixed SPC, GDO, and ethanol[31], as amphiphilic SPC could act as a protective sheath of IR820 and aPD-L1 to achieve complete dispersion in the oil phase. The mixed lipid formulation turned to gel, i.e., aPD-L1/I@LG, quickly when exposed to aqueous media (Fig. 2a). Moreover, the LG was able to experience a reversible gel-to-sol phase transition upon changing the temperature (Fig. 2a). Both shear storage modulus ($G'$) and shear loss modulus ($G''$) decreased with the increase of temperature, and the phase transition occurred at the temperature where the two moduli equaled to each other ($G' = G''$) (Fig. 2b). More interestingly, we can tune the phase transition temperature in a range of 29–67 °C by simply switching the mass ratio of SPC/GDO between 32/68 and 40/60 (Supplementary Fig. 1), suggesting that the LG is a flexibly adjustable formulation platform that perfectly meets the need of the all-in-control drug-delivery strategy. In the study presented here, we chose an SPC/GDO ratio of 35/65 to set the phase transition temperature at 39.5 °C. The LG modulus changes in three heating–cooling cycles between 37 °C and 43 °C were examined, which confirmed the reversible gel-to-sol transition of the LG within this temperature range (Fig. 2c). In addition, the phase transition behavior of the LG remained unchanged after loading IR820 and immunoglobulin G (IgG, 2.0 mg mL$^{-1}$) (Supplementary Fig. 2), suggesting that the encapsulated model drugs had no obvious influence on the mechanical property of the LG depot.

Furthermore, the LG degradation behavior was investigated both in vitro and in vivo. The in vitro gel degradation was mainly studied by determining the morphology and weight change of the formed gel in phosphate-buffered saline (PBS) (pH 7.4) in the presence of different lipase concentrations. It was found that the drug-loaded LG implemented a process of swelling followed by degradation (Fig. 2d and Supplementary Fig. 3) and the degradation became faster as the lipase concentration increased. Complete degradation of the LG could be observed on 6, 10, and 14 days with the respective lipase concentrations of 10%, 5%, and 2%. For the in vivo degradation test, the lipid precursor of the LG (50 µL) turned to gel within 1 h after subcutaneous injection into mice (Fig. 2e). A gel swelling process accompanied with volume increase was observed in the first 7 days after injection, although the degradation occurred concurrently. Afterwards, the gel volume gradually decreased in the following several weeks. It was notable that laser irradiation was able to accelerate the degradation of the gel. Finally, the LG was completely degraded within 1 month with laser irradiation, while a small gel matrix was still existing in the mice without laser irradiation (Fig. 2e). Furthermore, hematoxylin–eosin staining (H&E) images of the skins surrounding the injection site showed no chronic inflammatory reaction during the entire treatment procedure (Fig. 2e), indicating that the LG possessed a good biocompatibility for in vivo applications.

To further evaluate the stability and biocompatibility of the LG in vivo, IR820-loaded LG (IR820@LG) and a PBS solution of IR820 were injected into healthy mice, respectively. Monitored by

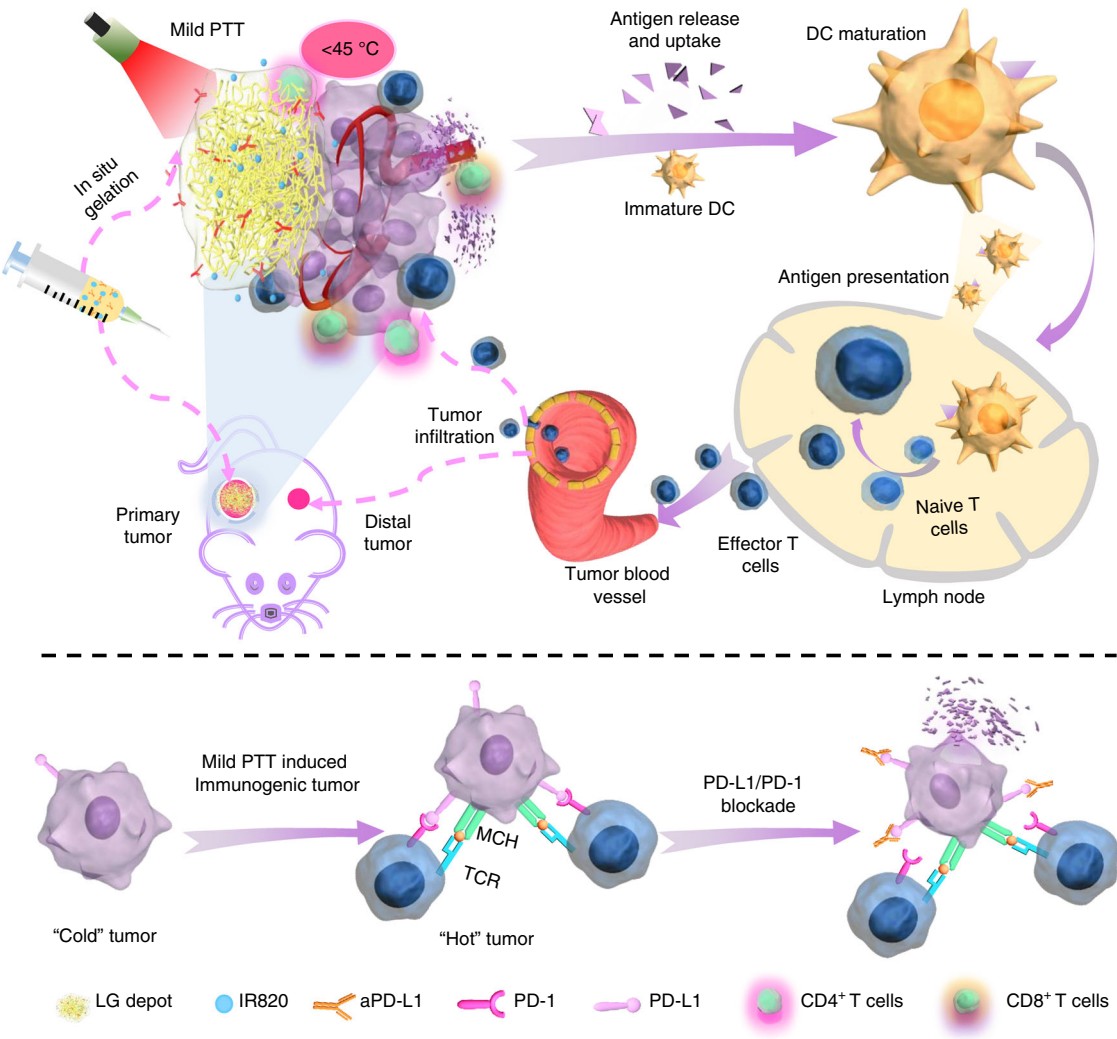

**Fig. 1** Schematic illustration of the symbiotic mild photothermal-assisted immunotherapy (SMPAI) via a combined all-in-one and all-in-control strategy. aPD-L1/I@LG can trigger a mild temperature rising and activate the systemic immunity to eradicate primary tumors and prevent tumor recurrence and metastasis

an in vivo imaging system (IVIS), the IR820@LG without irradiation showed the longest period of over 30 days in mice (Fig. 2f, g), suggesting that the LG was a stable platform for extended drug release. The fluorescence retention time of IR820@LG was reduced to around 25 days if the mice were irradiated by an 808 nm laser at 4 h and 48 h post injection, which indicated that photo-induced heat could also accelerate the degradation of the LG. In addition, 3-(4,5-dimethyl-thiazol-2-yl) −2,5-diphenyl tetrazolium bromide (MTT) studies on different cell lines exhibited a >80% cell viability in the tested groups that had been incubated with different concentrations of LG leachates for various time intervals, which confirmed the low cytotoxicity and excellent cytocompatibility of the LG (Supplementary Fig. 4).

**Photothermal effect and drug release behavior of IR820@LG in vitro and in vivo**. As is well known, IR820 is a photothermal and moderate photodynamic therapy agent[32,33]. To investigate the potential impact of photodynamic response on the therapeutic outcomes, we have compared the reactive oxygen species (ROS) production in IR820 PBS solution and IR820@LG using various ROS probes. As shown in Supplementary Figs. 5–7, NIR irradiation could only induce ROS production by IR820 in the solution. Interestingly, negligible amount of ROS was generated by IR820 in the LG depot, probably attributed to the extremely

hypoxia environment inside the LG. Moreover, the TME is usually considered hypoxia, which is also an adverse situation for ROS production. It should also be noted that the released IR820 could be easily eliminated from the tumor site during a long interval between two NIR irradiations. Thus, we can exclude the effect of IR820-induced photodynamic responses on tumor inhibition in the following in vivo antitumor studies.

We also examined the role of mild temperature. Briefly, after seeding and adherence, 4T1 cells were heated in a water bath at various temperatures, i.e., 37 °C, 40 °C, 43 °C, and 45 °C, for 10 min. MTT study was performed immediately or at 24 h after heating. As shown in Fig. 3a, the cell viability was not affected in different treatments, confirming that heating at 45 °C for 10 min was safe to the cells. In addition, western blot analysis was applied after 24 h of incubation post heat treatments, which showed that mild heating in the range of 37–45 °C could successively upregulate the expression of PD-L1 on the surface of the tumor cells (Fig. 3b). These results confirmed the rationality of our combination strategy with delivery of aPD-L1, which was responsible for sensitizing the tumor cells to ICB treatment and realizing immune normalization in the TME.

Next, the photothermal conversion efficiency and photothermal stability of IR820@LG in vitro and in vivo were explored, respectively. When irradiated by an 808 nm laser, the temperature

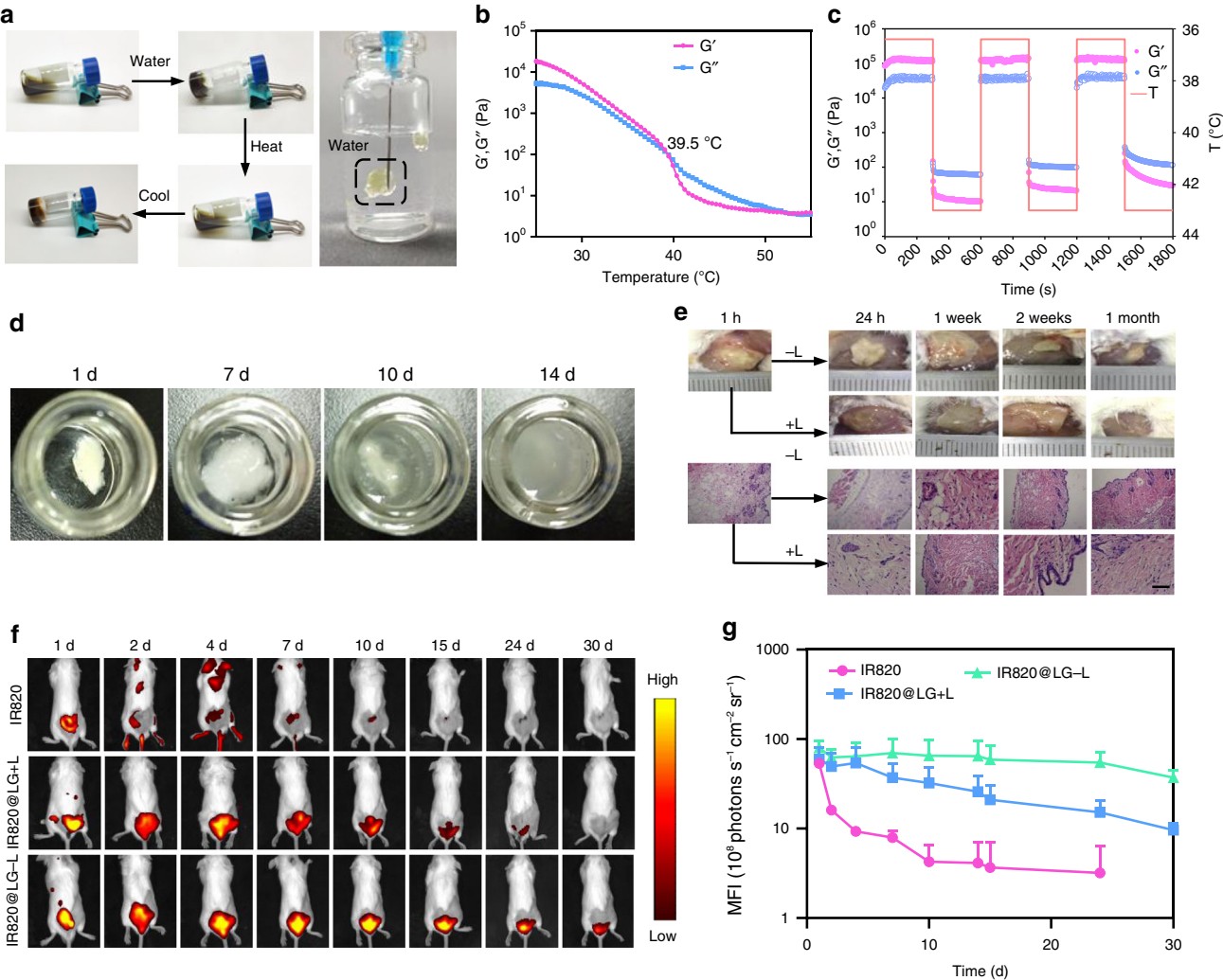

**Fig. 2** Function characterization of the LG depot. **a** Photographs of the sol-to-gel transition upon adding water and changing the temperature.
**b** Temperature-responsive storage ($G'$) and loss ($G''$) modulus changes of the LG with an SPC/GDO mass ratio of 35/65. The phase transition temperature of the LG is 39.5 °C. **c** Modulus changes of the LG with an SPC/GDO mass ratio of 35/65 with three thermal cycles of heating (43 °C) and cooling (37 °C).
**d** Morphology changes of the gel in 1 × PBS with 2% lipase over 14 d. **e** In vivo degradation behavior and tissue biocompatibility of the in situ-formed LG with H&E staining of the surrounding skin at different testing times (scale bar: 200 μm). **f** Fluorescence IVIS images depicting the in vivo retention of IR820 solution, IR820@LG with or without the NIR laser (808 nm). **g** Mean fluorescence intensity (MFI) of IR820 solution, IR820@LG with or without the NIR laser (808 nm) in non-tumor-bearing mice. Data are mean ± SEM ($n = 3$ biologically independent samples)

of both IR820@LG and a PBS solution of IR820 increased over 40 °C within a short irradiation time of 2 min. However, IR820@LG exhibited a much better photothermal stability than the IR820 solution in a multi-cycle irradiation study (Fig. 3c). The photo-induced temperature raising of IR820@LG kept constant during five consecutive on/off cycles of irradiation, whereas the IR820 solution exhibited a significant decay in photothermal conversion even after the first irradiation. This observation attested that LG can effectively conserve the photothermal conversion efficiency of the loaded photosensitizer. As a control, the temperature of blank LG or PBS did not increase after NIR irradiation (Supplementary Fig. 8). To achieve the desired mild heating, we first irradiated the mice injected with IR820@LG for a short time (around 2 min). After the temperature of the injected site reached 45 °C, the laser power was turned down and tuned manually to maintain this temperature for 10 min (Supplementary Fig. 9). We next investigated the intratumor photothermal conversion of IR820@LG. The infrared thermal images and temperature change of the tumor region in mice bearing with different tumor models (4T1 and B16F10) were recorded after

different formulations were injected intratumorally and irradiated. For both tumor models, prompt temperature increase on the tumor/injection site could be achieved for both IR820 PBS solution and IR820@LG when the tumors were irradiated by NIR at 4 h post injection, and the relatively low temperature of 45 °C could be maintained by manually tuning the laser power (Fig. 3d, e and Supplementary Fig. 10). However, when irradiating the mice at 48 h after being injected with the IR820 solution, the temperature around the tumor sites failed to reach 45 °C even after 10 min of irradiation. On the contrary, the mice injected with IR820@LG were still able to achieve a rapid temperature elevation when irradiated after 48 h post injection (Supplementary Fig. 11), suggesting that the LG can effectively preserve the photothermal conversion efficiency of IR820 for long-term SMPAI strategy.

We next examined the all-in-control release behavior of the drug-loaded LG triggered by manually controlled NIR irradiation both in vitro and in vivo. IgG were used as an alternative of aPD-L1 and co-loaded with IR820 into the LG to evaluate the reversible photo-responsive release profiles in vitro. IgG was

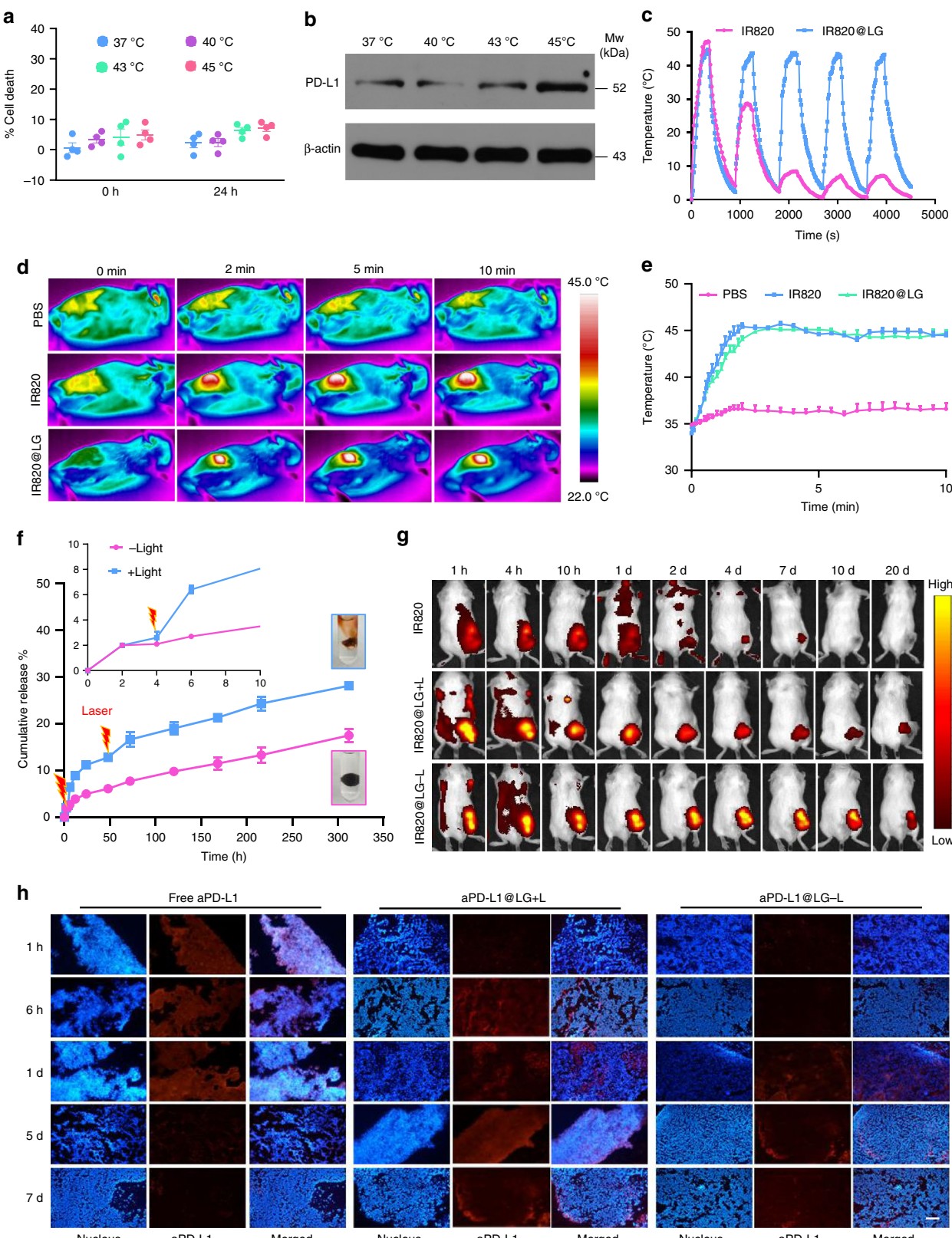

released from the LG sol form at an apparently higher rate than from the LG gel form (Supplementary Fig. 12). The release of IgG from IgG/IR820@LG was relatively stable without irradiation and 5% of IgG could be released from the formulation within 72 h. When NIR irradiation was applied at 4 h and 48 h, in a pattern similar to that shown in Fig. 3e, significantly enhanced release of

IgG could be triggered (Fig. 3f). Accordingly, over 15% of IgG was released from the LG at 72 h. To investigate the in vivo release behavior of the LG formulation triggered by NIR, IR820@LG and the PBS solution of IR820 were administered intratumorally into different groups of 4T1-tumor-bearing mice. From the IVIS images shown in Fig. 3g, the fluorescence decreased quickly after

**Fig. 3** Photothermal performance and drug release behavior of drug-loaded LG. **a** Cell death percentages of 4T1 tumor cells at 0 h and 24 h after heating at various temperatures, i.e., 37 °C, 40 °C, 43 °C, and 45 °C. Data are shown as means ± SEM ($n = 4$ biologically independent samples). **b** PD-L1 expression on 4T1 tumor cells after different treatments. **c** Heating and cooling curves of IR820@LG for five cycles by turning on and off laser. **d** In vivo infrared thermal images of the tumor sites in 4T1-bearing mice irradiated immediately post intratumor injection with PBS, IR820 solution, and IR820@LG. Images were recorded at 0 min, 2 min, 5 min, and 10 min after irradiation. **e** Temperature variation curves of the tumor sites in 4T1-bearing mice after intratumor injection with PBS, IR820 solution, and IR820@LG, followed by NIR laser irradiation. Data are shown as means ± SEM ($n = 3$ biologically independent samples). **f** Cumulative release profiles of IgG from LG incubated with PBS with or without 808 nm laser illumination. Data are shown as means ± SEM ($n = 3$ independent samples). **g** Fluorescence IVIS images describing the intratumor retention time of IR820 solution, IR820@LG with or without the NIR laser irradiation in 4T1 tumor-bearing mice. **h** Fluorescence images of tumors stained by Cy3-labeled second antibody (for aPD-L1) and DAPI (for nuclei). Tumors were treated with free aPD-L1, aPD-L1@LG with light (aPD-L1@LG + L), and aPD-L1 without light (aPD-L1@LG-L) at different time points (scale bar: 200 μm)

24 h post injection of the PBS solution of IR820, with a total intensity loss of 50% after 48 h (Supplementary Fig. 13). On the contrary, IR820@LG showed a prolonged fluorescence retention in tumors. The fluorescence intensity of the NIR-irradiated IR820@LG group decreased faster than that of the group without irradiation, possibly due to the photo-induced gel degradation and rapid IR820 loss. On the other hand, a dual-staining method was applied to investigate the intratumor release of aPD-L1. As shown in Fig. 3h, free aPD-L1 showed a strong red fluorescence in tumor immediately after injection, but the fluorescence diminished gradually and disappeared on day 7. In contrast, the initial fluorescence intensity was very weak in the aPD-L1@LG dosed group, showing the excellent inhibition against a burst release of aPD-L1. The fluorescence intensity of aPD-L1 kept on increasing consistently in the following 7 days and manually controlled NIR irradiation could accelerate the release of encapsulated antibody in the LG, indicating that this sol-to-gel phase transition triggered by NIR served as an excellent platform for effective on-demand drug release.

**In vivo validation of SMPAI strategy in 4T1 tumor mouse model**. We used mice bearing 4T1 tumors, which have been demonstrated to have low immune response with insufficient immunogenicity[29], to further validate whether the proposed SMPAI strategy can convert tumors from "cold" to "hot." The 4T1-tumor-bearing BALB/c mice were randomly grouped and each treated with a single intratumor injection of PBS, a mixed solution of aPD-L1 and IR820 (aPD-L1/I), IR820@LG, or aPD-L1/I@LG when the tumor volumes reached ~ 50 mm$^3$ on day 7. The dose of IR820 was 2.0 mg kg$^{-1}$, and that of aPD-L1 was 100 μg per mouse in each group. NIR irradiation with 808 nm laser was applied to the mice treated with IR820@LG, aPD-L1/I, or aPD-L1/I@LG on days 0, 2, 4, and 6 post injection in a pattern shown in Fig. 4a to achieve a mild PTT.

The combined mild PTT and immunotherapy showed an effective inhibition on the growth of primary tumors. The aPD-L1/I@LG + L group (namely the mice dosed with aPD-L1/I@LG and irradiated by NIR light) exhibited a tumor volume increase that was distinctly slower than all other groups and three mice in this group even exhibited no measurable tumors at the end of the study (Fig. 4b, d). The tumor growth on the mice injected with aPD-L1/I@LG but without NIR irradiation (the aPD-L1/I@LG-L group) was also restrained, but to a significantly less extent. However, it was noted that the IR820@LG + L group showed a similar tumor growth rate compared with the PBS control group, suggesting that mild PTT alone had a limited inhibition effect on the tumor growth. This observation is consistent with the fact that mild heat can upregulate the expression of PD-L1 and HSP in cancer cells and consequently promote the self-protection of tumor cells[32]. In addition, the primary tumors of two mice in the aPD-L1/I + L group disappeared after receiving the treatment but one of them recurred on day 13 (Fig. 4c), suggesting that non-

persistent antigen stimulation was insufficient to activate enough adaptive T-cell responses[33]. Attributed to the synergistic antitumor effect, the mice treated with aPD-L1/I + L and aPD-L1/I@LG + L significantly reduced the rate of tumor progression and resulted in a long-term survival rate of 17% and 50%, respectively (Fig. 4e). The above observations unambiguously evidenced the remarkable anticancer efficacy of the SMPAI strategy and the LG-based all-in-control strategy was necessary to maximize the synergistic effect of mild PTT and immunotherapy. In addition, the body weight showed negligible difference among all groups (Supplementary Fig. 14) and the H&E images confirmed no obvious damage and inflammation infiltration in the main organs, including the liver, lung, kidney, and heart (Supplementary Fig. 15), suggesting the safety of all formulations.

**SMPAI strategy turns "cold" tumors "hot"**. To verify the enhanced antitumor immune response induced by combined mild PTT and anti-PD therapy, we further measured the response of immune cells in lymph nodes, tumors, and spleens, as well as the major immune cell cytokines in the serum from a portion of the tested mice ($n = 3$) after the mice had been treated for 8 days (Fig. 5 and Supplementary Figs. 16–19). To our knowledge, dendritic cells (DCs) play a crucial role in initiating and regulating the innate and adaptive immunities[34]. Therefore, we first investigated if SMPAI strategy could promote DC maturation in the inguinal lymph nodes. As a result, the percentage of matured DCs (CD11c$^+$CD80$^+$CD86$^+$) in the aPD-L1/I@LG + L group (~42%) was significantly higher than that in the PBS group (23%). Correspondingly, the DC maturation percentages were ~38%, ~30%, and ~33% for the IR820@LG + L, aPD-L1/I@LG-L, and aPD-L1/I + L groups, respectively (Fig. 5a, b and Supplementary Fig. 16a). These results evidenced that SMPAI strategy could induce a strong immune stimulation effect. Meanwhile, we also identified the individual population of CD8$^+$ T cells and CD4$^+$ T cells in the primary tumors and spleen. CD8$^+$ T cells and CD4$^+$ T cells are both T lymphocytes and are crucial in the immune responses of antitumor therapy[35]. The populations of CD8$^+$ T cells and CD4$^+$ T cells in the aPD-L1/I@LG + L group were 11.8- and 8.2-folds higher than those in the PBS group (Fig. 5c, d and Supplementary Fig. 16b), indicating the effective infiltration of cytotoxic T cells into 4T1 tumors. In addition, the aPD-L1/I@LG-L group exhibited modest CD8$^+$ T-cell infiltration, which might be attributed to the inhibition of tumor-associated immunosuppression with a slow release of aPD-L1 from the LG. Markedly, the aPD-L1/I@LG + L group showed a 1.5- and 1.4-fold increase in tumor-infiltrating CD8$^+$ T cells compared with the aPD-L1/I@LG-L and IR820@LG + L groups, respectively, indicating that the SMPAI strategy could amplify the T cells immune response and sensitize the immunotherapy of "cold" tumors with relatively low level of TILs. In the spleen, the population of CD8$^+$ T cells and CD4$^+$ T cells in the aPD-L1/I@LG + L group were 3.6- and 6.4-fold higher than those in the

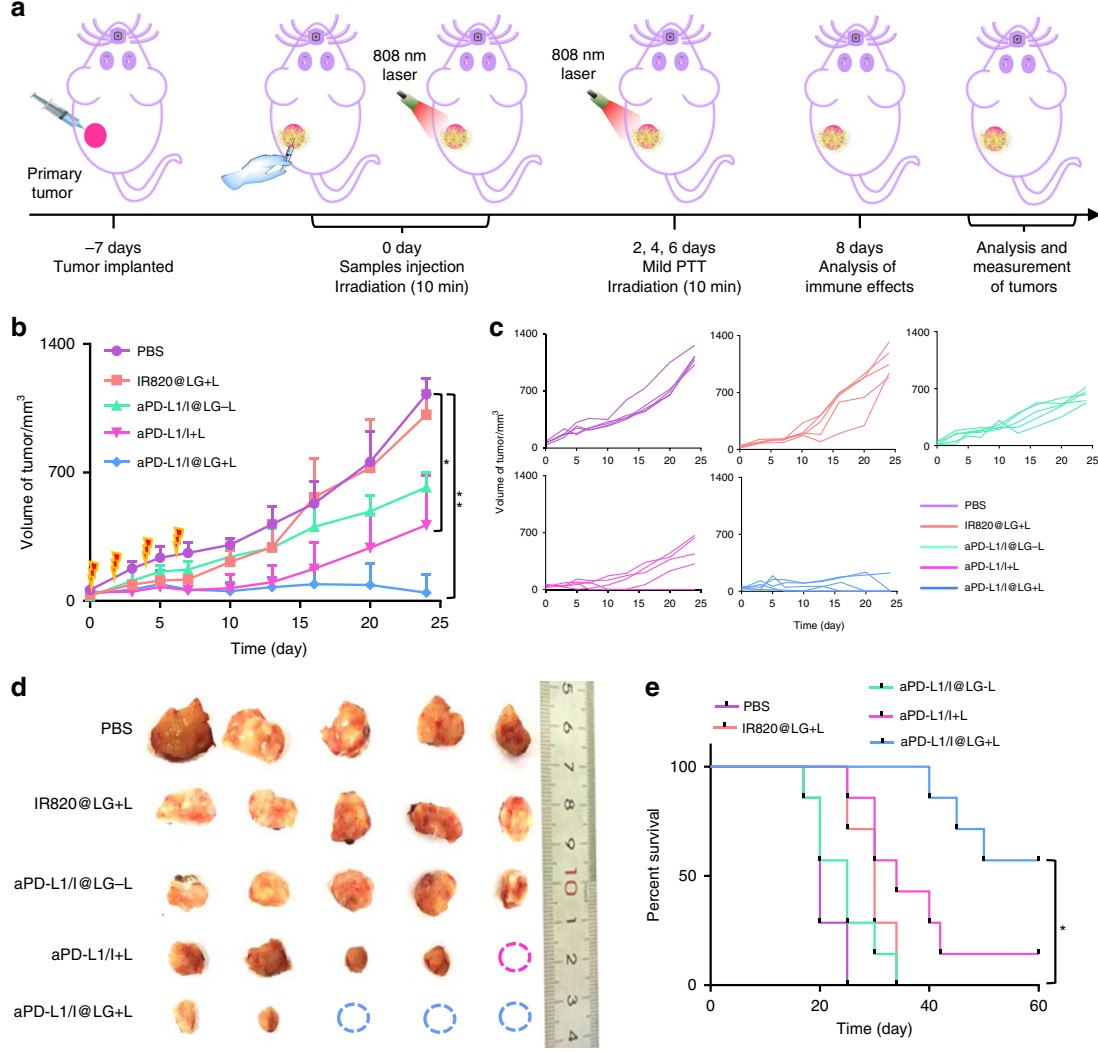

**Fig. 4** In situ gel depot for inhibition of 4T1 carcinoma tumor growth in vivo. **a** Schematic illustration of the animal experimental design. **b** Primary tumor growth curves with the mean tumor volumes of 4T1 tumor-bearing BALB/c mice model. Data presented as mean ± SEM (n = 5 biologically independent samples). **c** Primary tumor growth curves of individual mouse in different groups of 4T1 tumor-bearing BALB/c mice model. **d** The tumor images obtained from the tumor-bearing BALB/c mice on day 24 after treatment. **e** The survival percentages of the tumor-bearing BALB/c mice (n = 7 biologically independent samples). The comparison of two groups was followed by unpaired Student's t-test (two-tailed). *P < 0.05 and **P < 0.01

PBS group, and no significant difference in population of CD8+ T cells was observed among the other groups (Supplementary Fig. 17), indicating that this strategy-induced differentiation of naive T cells to CD8+ T cells that could help inhibit the growth of distant tumors and prevent tumor metastasis.

In addition, we examined the level of CD4+, CD25+, and Foxp3+ in tumor regulatory T cells (Tregs) by flow cytometry on day 8 after the start of treatment. The high expression of these proteins is an important symbol, indicating the suppression of Tregs to the antitumor immune responses of cytotoxic T lymphocytes (CTLs)[36–38]. The results demonstrated that the level of these proteins and the number of Tregs in the aPD-L1/I@LG + L group were much lower than those in the other groups, verifying significantly reduced tumor-associated immunosuppression in this group (Fig. 5e, f and Supplementary Fig. 16c). To further demonstrate that mild thermal environment could amplify aPD-L1-induced antitumor response, we also checked the frequency of myeloid-derived suppressor cells (MDSCs) in the tumors and spleens (Fig. 5g, h and Supplementary Fig. 18). As the decreased frequency of MDSCs stands for an enhanced systemic immune response, the aPD-L1/I@LG + L group had the lowest

MDSCs frequency, consistent with the effective suppression against the distant tumors in this group. The combination therapy did not cause any change in interleukin 6 (IL-6), which is secreted by T cells and macrophages to stimulate immune response (Supplementary Fig. 19). However, the aPD-L1/I@LG + L group showed significantly increased level of tumor necrosis factor α (TNF-α) and interferon-γ (IFN-γ) in the plasma (Fig. 5i, j), which are the key biomarkers released by immune cells of the TME for altering T-cell responses[39]. These results indicated that the local SMPAI strategy via the all-in-one and all-in-control strategy can sensitize immunologically "cold" tumors for amplified anti-PD immune therapy, as well as stimulate the systemic immune response for effective inhibition of tumor metastasis.

**Inhibition of the distal tumors growth by SMPAI strategy.** We have also examined whether the established active immune responses were strong enough to inhibit an untreated distal tumor. In this study, the distal tumor model was established simultaneously with the primary tumors (Fig. 6a). After local treatment on primary tumors, the size change of the distal tumors

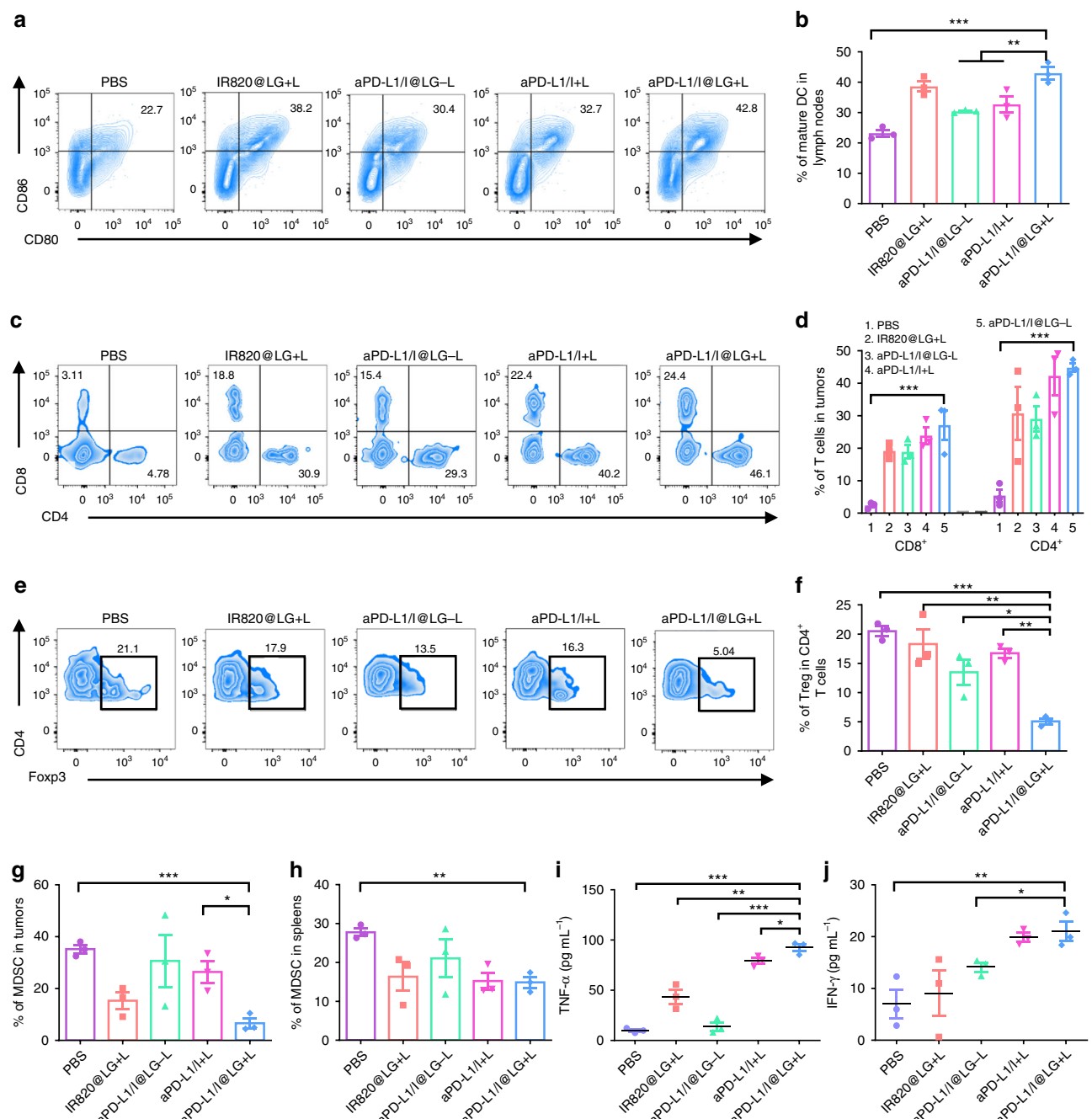

**Fig. 5** SMPAI strategy-induced immune responses. **a**, **b** DC maturation induced by SMPAI strategy on 4T1 tumor-bearing mice (gated on CD11+ DC cells). Cells in the lymph nodes were collected on day 8 after various treatments for assessment by flow cytometry after staining with CD11c+, CD80, and CD86. Data represent mean ± SEM ($n = 3$ biologically independent samples). **c**, **d** Flow cytometric examination of the intratumor infiltration of CD4+ and CD8+ T cells (gated on CD3+ T cells). Data represent mean ± SEM ($n = 3$ biologically independent samples). **e**, **f** The Treg frequencies in tumors after different treatments examined on day 8 after treatment. Data represent mean ± SEM ($n = 3$ biologically independent samples). **g**, **h** The MDSC frequencies in tumors and spleens after different treatments examined on day 8 after treatment. Data represent mean ± SEM ($n = 3$ biologically independent samples). **i**, **j** Contents of the TNF-α and IFN-γ in plasma on day 8 after treatment. Data represent mean ± SEM ($n = 3$ biologically independent samples). The comparison of two groups was followed by unpaired student's t-test (two-tailed). $*P < 0.05$, $**P < 0.01$, and $***P < 0.001$

was recorded and plotted (Fig. 6b, c). Strikingly, the growth of distal tumors was significantly slowed down on the mice treated with formulations containing aPD-L1, i.e., the aPD-L1/I@LG-L, aPD-L1/I + L, and aPD-L1/I@LG + L groups, which confirmed that aPD-L1 played a key role in the systemic immunity establishment, as well as the distal tumor inhibition. However, the outcome of single treatment, i.e., IR820@LG + L or aPD-L1/I@LG-L, was limited on the distal tumors, suggesting that

systemic immunity was likely restricted in the immunosuppressive TME of the distal tumors without a combined therapy. In addition, the aPD-L1/I@LG + L group achieved much better efficacies on distal tumors than the aPD-L1/I + L group, indicating that the sustained release of antibodies from the LG depot could maintain a long-term abscopal immune response, which was critical to rebuild the "cold" TME and realize ideal ICB outcomes.

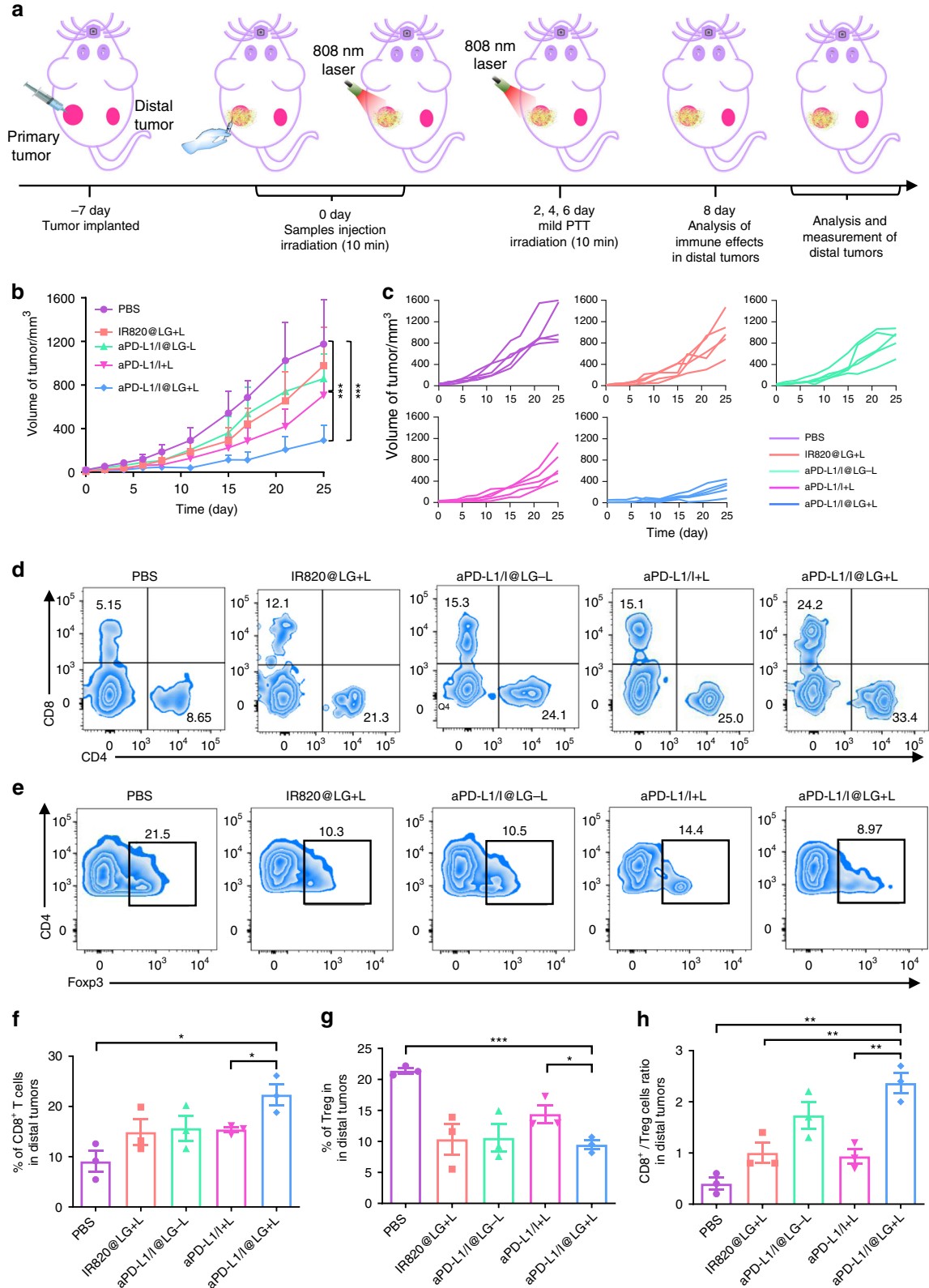

To better understand the improved tumor inhibition effect of the SMPAI strategy on distal tumors, the intratumor infiltration of CD8[+] CTLs and Tregs were examined by flow cytometry. The frequency of CD8[+] CTLs infiltrating in distal tumors of the aPD-L1/I@LG + L group was 4.7- and 1.6-fold higher than those of the PBS group and the aPD-L1/I + L group,

respectively (Fig. 6d, f). Meanwhile, Treg frequency in the aPD-L1/I@LG + L group was much lower than in the PBS group, verifying that the SMPAI strategy significantly reversed the immune tolerance of the distal tumors (Fig. 6e, g). In addition, the CD8[+]/Treg ratio, a critical parameter to evaluate the antitumor immune responses, was greatly elevated in the

**Fig. 6** The therapeutic effect on abscopal tumors with SMPAI strategy. **a** Schematic illustration of the animal experimental design for distal tumors. **b** Distal tumor growth curves with the mean tumor volumes of 4T1 tumor-bearing BALB/c mice model ($n = 5$ biologically independent samples). **c** Distal tumor growth curve of individual mouse in different groups of 4T1 tumor-bearing BALB/c mice model. **d** Representative flow cytometry plots showing different groups of T cells in distal tumors (gated on CD3+ T cells). **e** Representative flow cytometry plots showing percentages (gated on CD4+ cells) of Tregs in distal tumors on day 8 after different treatments. **f** Proportions of the intratumor infiltration of CD8+ killer T cells in distal tumors. Data represent mean ± SEM ($n = 3$ biologically independent samples). **g** The frequency of Tregs in distal tumors upon various treatments. Data represent mean ± SEM ($n = 3$ biologically independent samples). **h** CD8+ CTL/Treg ratios in the distal tumors after various treatments. Data represent mean ± SEM ($n = 3$ biologically independent samples). The comparison of two groups was followed by unpaired Student's $t$-test (two-tailed). *$P < 0.05$, **$P < 0.01$, and ***$P < 0.001$

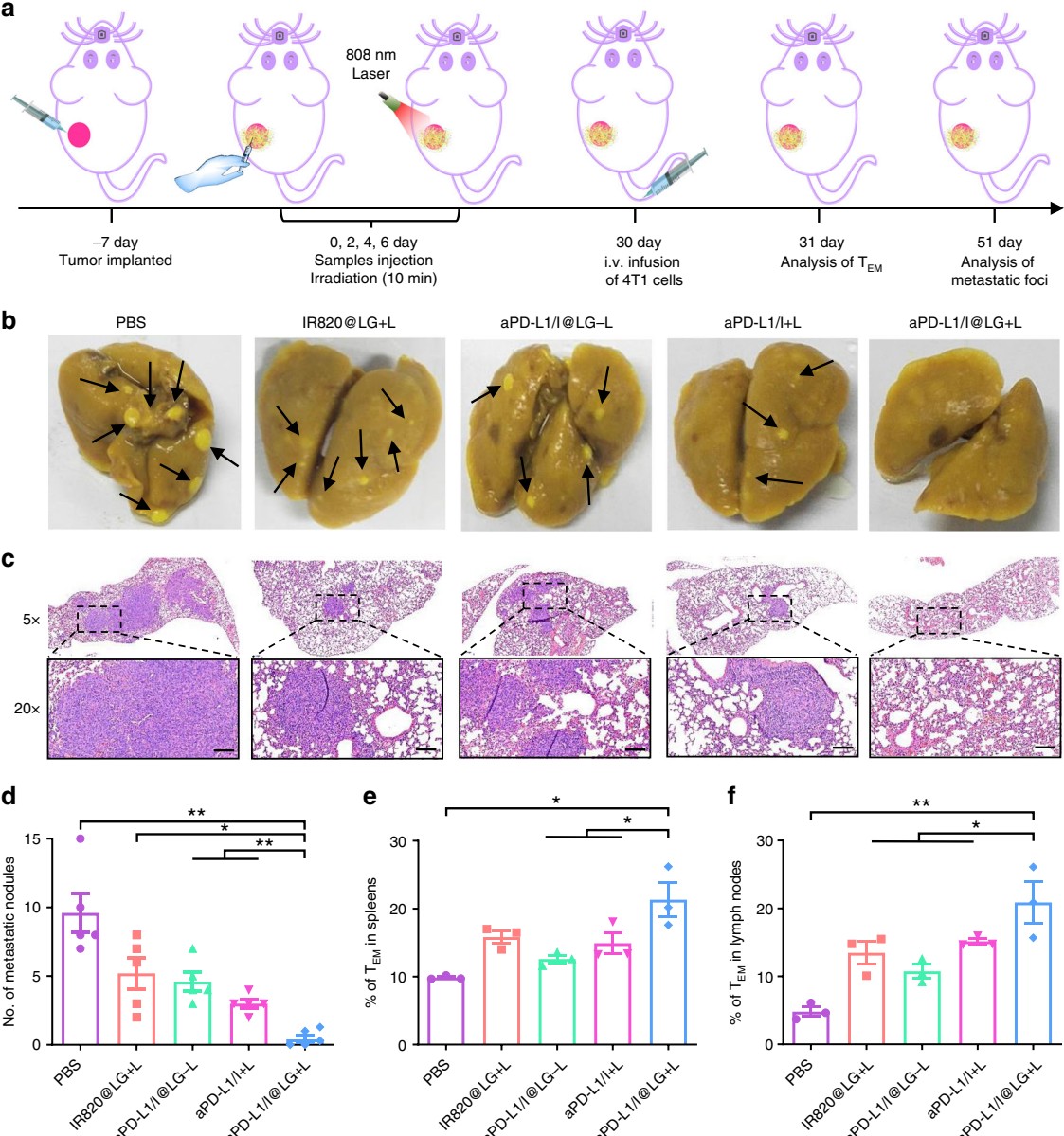

**Fig. 7** Metastasis prevention via SMPAI-induced long-term immune effects. **a** Schematic illustration for SMPAI-mediated inhibition of tumor metastasis. **b** Representative photographs of lung tissues with tumor metastasis. **c** H&E staining of lung tissues collected on day 51 (scale bars: 100 μm). **d** Quantification of pulmonary metastasis nodules in different groups of 4T1 tumor-bearing BALB/c mice ($n = 5$ biologically independent samples). **e, f** Proportions of $T_{EM}$ cells in the spleens and lymph nodes (gated on CD3+ CD8+ T cells) examined after 1 day post intravenous infusion of the 4T1 cells. Data are presented as mean ± SEM ($n = 3$ biologically independent samples). The comparison of two groups was followed by unpaired Student's $t$-test (two-tailed). *$P < 0.05$ and **$P < 0.01$

distal tumors of the mice treated with aPD-L1/I@LG + L rather than in the other groups (Fig. 6h).

**Long-term immune memory effects**. Encouraged by the excellent performance of the SMPAI strategy in inhibiting tumor recurrence and distal tumors, we further evaluated the therapeutic efficacy of our strategy in a more aggressive whole-body spreading tumor model. In this study, mice were intravenously injected with 4T1 cells on day 30 after the start of various treatments (Fig. 7a). After feeding the mice for another 21 days, the lung tissues in different groups were collected for metastasis analysis. As shown in Fig. 7b, c, lung metastatic foci were found in all groups, except in the aPD-L1/I@LG + L group, suggesting the success of our SMPAI strategy in inhibiting the lung metastasis. In addition, significantly more pulmonary metastasis nodules were observed in all control groups other than in the aPD-L1/I@LG + L group (Fig. 7d). These results further declared that the LG-based all-in-control strategy was beneficial for the establishment of active immune responses to the metastatic 4T1 tumors.

We further analyzed the adaptive immunity establishment at cellular levels. In general, central memory T cells ($T_{CM}$ cells) provide protections after antigen-stimulated clonal expansion, differentiation and trafficking, whereas effector memory T cells ($T_{EM}$ cells) can induce immediate protections by producing cytokines, such as IFN-γ[40–42]. Therefore, to understand the mechanism underlying the tumor-specific anti-metastasis property of the SMPAI strategy, the $T_{EM}$ cells in the spleens and lymph nodes were examined using flow cytometric measurement on the next day of intravenous infusion of 4T1 cells. The mice treated with aPD-L1/I@LG + L showed significantly increased $T_{EM}$ cell frequency in the spleens and lymph nodes, whereas all the other groups exhibited a moderate generation of $T_{EM}$ cells (Fig. 7e, f and Supplementary Fig. 16d), suggesting that the SMPAI strategy could produce a better memory immunity effect that was helpful to inhibit cancer metastasis.

**In vivo validation of SMPAI strategy in B16F10 mouse tumor model**. To further demonstrate that our proposed SMPAI strategy is broadly applicable in treating immunologically "cold" tumors, malignant B16F10 melanoma tumor[30], another mouse tumor model with scarce T-cell infiltration in TME, was employed for in vivo studies. We first inspected the retention time of aPD-L1/I@LG in the B16F10 melanoma tumors. The LG was almost completely degraded within 10 days, much shorter than the retention time in 4T1 tumor model (Fig. 8a, b). Typically, gels are corroded more easily in the B16F10 tumors than in other types of solid tumors after intratumor administration because of the high plasticity of B16F10 tumors with vast blood flow[43]. As a result, faster drug release from LG would be expected in B16F10 tumors than in 4T1 tumors. Therefore, reduced NIR irradiation time were applied in the melanoma treatment. For in vivo antitumor study, B16F10 cells were inoculated subcutaneously into the right flank of female C57BL/6 mice. Similar to the study on the 4T1 model, the tumor growth in the aPD-L1/I@LG + L group was significantly restrained after a scheduled NIR irradiation on day 0 and 4 post injection (Fig. 8c). As shown in Fig. 8d, e, the aPD-L1/I@LG-L group only showed a slight tumor inhibition effect, possibly due to the insufficient cytotoxic T cells in the TME. The great performance of aPD-L1/I + L against tumor in the first several days after injection suggested that sufficient mild heating was important. The persistent release of aPD-L1 and immunity stimulation by the aPD-L1/I@LG + L formulation was essential to achieve an enhanced tumor inhibition and a prolonged mouse survival rate (Fig. 8f). In addition,

the body weight profiles of all groups did not reflect significant difference, showing that the treatment did not cause obvious toxicity (Supplementary Fig. 20).

Furthermore, lymphocytes in the spleens and tumor tissues were analyzed by flow cytometry on day 8 after the start of treatment. The aPD-L1/I@LG + L group exhibited the highest population of CD8+ T cells infiltration in both the tumor site and spleen (Fig. 8g, j), indicating the upregulation of cytotoxic T-cell infiltration in the TME and the establishment of systemic immune responses as well. Therefore, the SMPAI strategy have been demonstrated to effectively increase the recruitment of TILs and sensitize immunologically "cold" tumors. In addition, the aPD-L1/I@LG + L group also exhibited the highest CD8+ T cells population in the spleen, which was an essential indicator to reflect the establishment of systemic immune response. These results suggested that the photo-responsive in situ LG was able to serve as a universal all-in-one and all-in-control depot to treat different types of cancers that are immunologically "cold".

## Discussion

Beyond the all-in-one strategy, we demonstrate an all-in-control system to emphasize the importance of personalized cancer therapy. In our opinion, an optimal scenario for clinical cancer therapy should be based on the comprehensive consideration of cancer type, stage, size, microenvironment, etc. An ideal drug-delivery system should be well designed to balance all factors as well as their responses[44–48]. In our case, the LG depot has a very flexible formula and can be manufactured through a facile preparation procedure[49]. By easily adjusting the parameters, such as the mass ratio of SPC/GDO, the drug loading of each payload, and the irradiation strength, we can achieve designated drug-release behavior and antitumor effect. In detail, the gel strength can be controlled by tuning the SPC/GDO mass ratio to achieve a consistent retention time in different tumors; the exposure dose can be controlled by adjusting the drug loading to maintain a minimum but effective concentration at focus; the drug release amount and schedule can be regulated by controlling gel-to-sol phase transition to realize constant antitumor effects. Thus, rationally designed LG depot can be regarded as a promising platform to realize personalized cancer therapy.

On the basis of these benefits, we have developed a SMPAI strategy based on a combined all-in-one and all-in-control gel depot for the treatment of immunologically "cold" tumors. The co-loaded IR820 and aPD-L1 in the thermal-responsive LG depot enabled persistent execution of concurrent mild PTT and immunotherapy. Manually controlled NIR irradiation was able to tune the reversible gel-to-sol phase transition of LG for the programmed release of aPD-L1 and to enhance the infiltration of T cells into the tumor, thereby increasing the potential for a response to ICB. This strategy has successfully rebuilt the "cold" TME for amplified anti-PD immune therapy against both 4T1 and B16F10 mouse models. The long-term retention of drug depot at focus fundamentally addressed the issue of rapid diffusion and metabolism of locally injected free drugs. In addition, a satisfactory immunotherapeutic effect depends not only on the efficient delivery of drugs but also on the status of tumors. Based on recent studies, a "cold" tumor is not sensitive to immune checkpoint inhibitors due to its low immunogenicity, which could negatively influence the tumor recognition capability of cytotoxic T cells[50,51]. In the current study, the precise control of mild PTT temperature is not only important to adjust the release of drugs but is also critical in sensitizing the immunodeficient tumors as well as potentiating the anti-PD-L1 treatment. Therefore, in future studies, it is essential to establish more complete temperature–effect relationships in vivo for optimized therapeutic

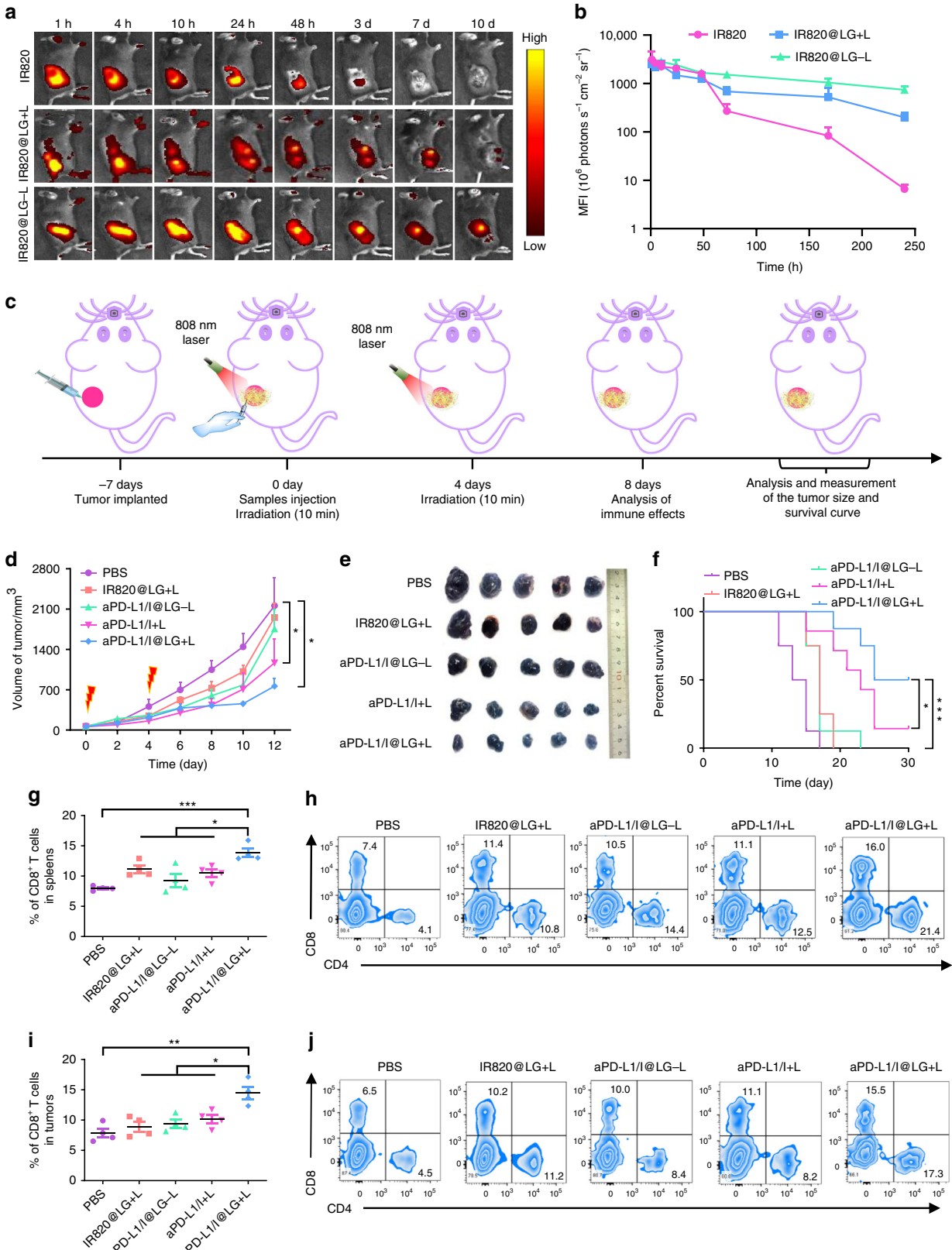

efficacy. Above all, the primary tumors receiving treatments exhibited a great shrinkage in volume and the growth of distal 4T1 tumors were also effectively restrained. This in situ SMPAI strategy based on the thermal-responsive LG holds promise as a localized drug-delivery platform for enhancing the therapeutic efficacy of ICB-based immuno-oncology.

## Methods

**Materials**. SPCs (Lipoid S100) were obtained from Lipoid GmbH (Ludwigshafen, Germany) with a purity of 97.6%. It was used as received and stored at −20 ℃. GDO was a kind gift from Croda, UK, which contained minimally 95% diglycerides according to the producer. All tool compounds were used as obtained. IR820 was purchased from Sigma. aPD-L1(catalog number BE0101) used in vivo purchased from Bio X Cell. Anti-CD3-PerCP-Cy5.5 (catalog number 551163), anti-CD4-

**Fig. 8** SMPAI strategy for the treatment of B16F10 growth in vivo. **a** Fluorescence IVIS imaging describing the intratumor retention time of IR820 solution, IR820@LG with or without the NIR laser (808 nm) in B16F10 tumor-bearing mice. **b** MFI of the intratumor drug retention time of IR820@LG with 808 nm laser in B16F10 tumor-bearing mice. The experiment was performed with $n = 3$ biologically independent samples. **c** Schematic illustration of the animal experimental design on B16F10 tumor-bearing mice. **d** The tumor growth curves of B16F10-bearing C57BL/6 mice model. Data are shown as mean ± SEM ($n = 5$ biologically independent samples). **e** The tumor images obtained from the tumor-bearing C57BL/6 mice at 12th day after treatment. **f** The survival percentages of the B16F10-bearing C57BL/6 mice ($n = 4$ biologically independent samples). **g**, **h** The proportions and flow cytometry plots of CD8$^+$ T cells in the spleens examined on day 8 after treatment. Data represent mean ± SEM ($n = 4$ biologically independent samples). **i**, **j** The proportions and flow cytometry plots of CD8$^+$ T cells in the tumors examined on the 8th day after treatment. Data are shown as mean ± SEM ($n = 4$ biologically independent samples). The comparison of two groups was followed by unpaired Student's $t$-test (two-tailed). *$P < 0.05$, **$P < 0.01$, and ***$P < 0.001$

FITC (catalog number 553046), anti-CD8-PE (catalog number 553032), anti-CD4-FITC (catalog number 553046), anti-CD25-APC (catalog number 557192), and anti-Foxp3-PE (catalog number 563101), anti-CD11b-FITC (catalog number 557396), anti-CD11c-FITC (catalog number 557400), anti-CD80-PE (catalog number 560016), anti-CD86-APC (catalog number 553692), anti-LY-6G/LY/6C-PE (catalog number 553128), anti-CD3-FITC (catalog number 553065), anti-CD8-PerCP-Cy5.5 (catalog number 553030), anti-CD62L-APC (catalog number 562910), and anti-CD44-PE (catalog number 559250) were purchased from BD Biosciences.

**Cell lines**. The fibroblast NIH 3T3, human blood B lymphocytes RAMOS (RA1), melanoma B16F10, and metastatic murine 4T1 breast cancer cell lines were purchased from the American Type Culture Collection. The NIH 3T3 cells were cultured in complete Dulbecco's modified Eagle's medium (DMEM; Gibco, Invitrogen) with 10% fetal bovine serum (FBS), penicillin (100 U mL$^{-1}$), streptomycin (100 U mL$^{-1}$), and 1% L-glutamine. The RA1 and 4T1 cells were maintained in Roswell Park Memorial Institute (RPMI) 1640 (Gibco, Invitrogen) medium with 10% FBS, penicillin (100 U mL$^{-1}$), streptomycin (100 U mL$^{-1}$), and 1% L-glutamine.

**Animals**. BALB/c mice and C57BL/6 mice (6–8 weeks old, 18–20 g) were purchased from the Qinglongshan Farms (Nanjing, China). All animals were bred in the pathogen-free facility with a 12 h light/dark cycle at 20 ± 3 °C and had ad libitum access to food and water. Animal protocols were performed under the guidelines for human and responsible use of animals in research set by Huazhong University of Science and Technology and China Pharmaceutical University.

**Preparation of the aPD-L1/I@LG**. All sample compositions herein were given as percentages by weight, unless otherwise stated. The drug-loaded LG samples were fabricated via the "Macrosol" technique according to the reported previously[31]. Briefly, 100 mg of SPC and 2.0 mg of IR820, or 1.0 mg of IgG, or 2.0 mg of aPD-L1 were added into 0.5 mL of pure water. The mixture was mixed sufficiently and lyophilized. The oil phase precursor preparation of LG was prepared via mixing appropriate amounts of lipids (SPC/GDO, 35/65) and ethanol (10%). Finally, 1.0 g of the precursor preparation was introduced into the lyophilized powder system mentioned above. The non-aqueous mixtures were then mixed on a rolling mixer at room temperature until homogenous lipid solution formulations were produced. The LG preparation was done under sterile conditions.

**The sol-gel phase test**. Two hundred milligrams of the precursor preparation of LG was moved into little glass bottle vials with cover. Then, the sol-gel transfer behavior was characterized by the test tube inverting method through adding a certain amount of water. The reversibly temperature response was investigated by raising and lowering the temperature from 25 °C to 45 °C.

**Rheological test**. Rheology experiments were performed using a dynamic shear rheometer (Kinexus Rotational Rheometer, Malvern Instruments, Malvern, UK). Three hundred milligrams of all-prepared LG was placed on the middle of a 15 mm diameter parallel plate with a proper gap and equilibrated the sample at 25 °C for 5 min before starting each measurement. Temperature ramp experiments were conducted within the range of 25–80 °C to study its thermo-sensitive sol-gel transition behavior, with a heating rate of 1 °C min$^{-1}$. A temperature cyclic step tests between 37 °C and 43 °C was also carried out, with angular frequency (ω) and strain (γ) held constant at 2 rad/s and 5%, respectively. To prevent the evaporation of water, a lid was prepared on the top.

**LG degradation behaviors**. One hundred milligrams of the LG (SPC/GDO, 35/65) with 0%, 2%, 5%, and 10% lipase were placed in the vials with an inner diameter of 5.0 cm. The phosphate buffer solution (PBS) at pH 7.4 were used as the degradation media. All the samples were incubated at 37 °C with gentle shaking. The morphology changes of LG were recorded by photographing. Furthermore, the gel remaining mass of each sample without the media was recorded by weight in the designed time intervals.

**Cytotoxicity evaluation in vitro**. Two hundred milligrams of the LG was extracted using DMEM for 24 h. Sequential dilutions of the stock solution were prepared to vary the concentrations of the leachates. The NIH 3T3, B16F10, and RA1 cell lines were seeded in 96-well plates with overnight incubating in DMEM. One hundred microliters of LG leachates at different concentrations, i.e., 100%, 50%, 25%, 12.5%, and 6.25%, were added in each well and incubated for another 24 and 48 h. Then, the cell viability was evaluated by MTT assay. Each data point was measured three times.

**IgG release from the LG**. Two hundred milligrams of IgG/IR820@LG (IgG, 1.0 mg mL$^{-1}$; IR820, 2.0 mg mL$^{-1}$) was added into each vial. The samples were divided into two groups (light group and non-light group) and were incubated on an shaker at different temperature (37 °C and above the phase transition temperature). Furthermore, 2.0 mL of PBS (pH 7.4) was used as the drug-release medium. At the desired interval, all the media of the sample was collected and stored at −20 °C for further analysis and another 2.0 mL of fresh medium was then added to the vial. Each sample of the light group was irradiated with an 808 nm laser at a power density for 10 min at 4 h and 48 h after sampling. The released amount of IgG was measured by a micro BCA protein assay kit (Boster) and diluted free IgG as a standard curve.

**In vivo evaluation of LG degradation**. In vivo degradation of IR820@LG was monitored using the IVIS Spectrum In Vivo Imaging System (IVIS Lumina XR, USA) after the IR820@LG and IR820 solution were injected into mice. In addition, each mouse of light group was then irradiated with an 808 nm laser for 10 min at an appropriate power density for four (4T1 tumor model) or two times (B16F10 tumor model) post injection. Fluorescence imaging was obtained at the desired interval and was analyzed by a Living Imaging software.

**In vitro and in vivo photothermal effect**. To investigate the photothermal stability of IR820-loaded LG, 200 mg of IR820@LG (IR820, 0.2 mg mL$^{-1}$) placed in a 2 mL tube were irradiated for five cycles with an 808 nm laser on and off at a power density of 1.0 W cm$^{-2}$. The heating and cooling curves of IR820-loaded LG were recorded by an infrared thermal camera (FLIR E50, USA). Furthermore, free IR820 dissolved in PBS was used as a control treated in the same conditions. In addition, 200 mg of IR820@LG, free IR820 solution, PBS, and blank LG placed in a 2 mL tube were irradiated with an 808 nm laser for 10 min at a lower power density for keeping the temperature below 45 °C. The temperature increase and thermal images of each samples were recorded.

In vivo Infrared Thermal Imaging was established as described above. The tumor was grown by inoculating 100 μL of 4T1 cells or B16F10 cells with a concentration of $1 \times 10^6$ cells in PBS into the right flank of each BALB/c mouse (female, 6 weeks old, 18–20 g) or C57BL/6 mouse (female, 6 weeks old, 18–20 g) under anesthesia.

*In vivo Infrared Thermal Imaging*. Tumor model was established as described above. After the tumor volume reached about 50 mm$^3$, 50 μL of IR820@LG precursor or free IR820 solution was injected directly into the tumor area. The tumor area was then irradiated with an 808 nm laser at an appropriate power density for 10 min. The thermal image of the whole mouse was recorded by the infrared thermal camera at 4 h and 48 h post injection. PBS and blank LG were used as a control treated in the same conditions.

**Intratumor drug retention test**. To evaluate the intratumor drug retention of IR820 (a model fluorescent molecule) and aPD-L1, the gels were injected into the tumors mildly. For controls, IR820 solutions were administered into the tumor sites. Fluorescence was monitored using the IVIS Spectrum In Vivo Imaging System (IVIS Lumina XR, USA). Then, the tumors were collected and stored by freezing at different intervals, respectively. At the last time point, all tumor frozen sections were obtained and stained by DAPI- and Cy3-labeled second antibody, which labeled the nucleus and aPD-L1, respectively, then the fluorescence images were collected in a confocal laser scanning microscope (Zesis 710, Germany).

**Antitumor study on 4T1 tumor models and B16F10 tumor models**. To evaluate the short-term efficiency of a depot of photothermal and immune checkpoint

inhibitor for synergistic cancer therapy, the antitumor study was performed using a 4T1 tumor model and B16F10 tumor model. For the tumor inoculation, 4T1 cells or B16F10 cells ($1 \times 10^6$) suspended in PBS were subcutaneously injected into the left flank of each female BALB/c or C57BL/6 mouse, respectively. The therapy started when the primary tumor volumes reached 50~100 mm$^3$. The 4T1 tumor-bearing mice were divided into five groups randomly and administrated with 50 μL of PBS, free aPD-L1/I solution, IR820@LG + L, aPD-L1/I@LG-L, and aPD-L1/I@LG + L (aPD-L1, 2.0 mg mL$^{-1}$; IR820, 2.0 mg mL$^{-1}$) through intratumor injection. In addition, each mouse in the light group was then irradiated with an 808 nm laser at an appropriate power density for keeping the temperature below 45 °C for four (4T1 tumor model) or two times (B16F10 tumor model) post injection. Then the tumor volume and body weights of mice were recorded from the first day of treating until the end of the experiment. The tumor volume was calculated as the following formula: width$^2$ × length × 0.5. On day 24, five other mice were killed to take photos. Finally, the survival time of the remaining mice for each group were monitored until day 60 post the first administration and the survival curves were produced. In addition, all the major organs (i.e., heart, liver, lung, and kidney) were collected with the examination by H&E staining.

**Inhibition of distal tumor growth**. To evaluate the abscopal therapeutic effect for distal tumors, the inhibition of tumor growth study was performed using a 4T1 tumor model. For the primary tumor inoculation, 4T1 cells or B16F10 cells ($1 \times 10^6$) suspended in PBS were subcutaneously injected into the left flank of each female BALB/c or C57BL/6 mouse. The distal tumors were inoculated simultaneously with the primary tumors by injecting 4T1 cells ($5 \times 10^5$) suspended in PBS into the right flank of each female BALB/c mouse. After the tumor volume reached about 50~100 mm$^3$, mice were divided into five groups ($n = 5$) and were treated as described above. Then the distal tumor volume of mice was recorded from the first day of treating until the end of the experiment.

**Anti-metastasis effect**. To establish the lung metastases tumor model, 4T1 cells ($1 \times 10^6$) suspended in PBS were subcutaneously injected into the left flank of each female BALB/c mouse. When the tumor reached 50~100 mm$^3$, the mice were randomly divided into five groups ($n = 8$ in each group) and were treated with the desired formations and selectively irradiated with 808 nm laser. The treatment was repeated for four times at an interval of 7 days. Thirty days later, each mouse was intravenously infused with $1 \times 10^5$ 4T1 tumor cells. Three mice were killed to obtain their spleens and lymph nodes for analysis of memory T cells on day 31. The remaining mice were killed on day 51 and lungs were fixed in Bouin's solution. Tumor metastasis sites subsequently appeared as yellow nodules on the surface of the lungs and were counted under a microscope. The establishment of 4T1 tumors in the lung was also examined by H&E staining.

**Cytokine detection**. Serum samples were isolated from mice on day 8 after various treatments and diluted for analysis. TNF-α, IFN-γ, and IL-6 were analyzed with ELISA kits according to vendors' instructions (KeyGEN Biotech, China).

**Ex vivo analysis of different groups of immune cells**. To examine the immune response caused by the combinational therapy, the primary tumors, distal tumors, lymph nodes, and spleens were surgically resected from mice in different groups. Then lymphocytes in the spleen and infiltrating lymphocytes in tumors were obtained after several operation. The collected lymphocytes were incubated with anti-CD3-PerCP-Cy5.5 (BD), anti-CD4-FITC (BD), and anti-CD8-PE (BD) antibodies according to the standard protocols, to determine the content of CD4$^+$ or CD8$^+$ T cells in the tumors and spleens using a flow cytometry. The Tregs was examined by staining the lymphocytes with anti-CD4-FITC (BD), anti-CD25-APC (BD), and anti-Foxp3-PE (BD) antibodies according to the standard protocols. The MDSCs were examined by staining the lymphocytes with anti-CD11b-APC (BD) and anti-LY-6G/LY6C-PE antibodies according to the standard protocols. For DC maturation examination in vivo, the inguinal lymph nodes were harvested to collect lymphocytes. The frequency of matured DCs in the lymph nodes was then examined by flow cytometry (BD FASCVerse) after immunofluorescence staining with anti-CD11c-FITC (BD), anti-CD80-PE (BD), and anti-CD86-APC (BD) antibodies according to the procedure of the manufacturer. For analysis of memory T cells, bone marrows collected from mice after various treatment were stained with anti-CD3-FITC (BD), anti-CD8-PerCP-Cy5.5 (BD), anti-CD62L-APC (BD), and anti-CD44-PE (BD) antibodies according to the manufacturer. The single-cell suspension from spleens and lymph nodes was prepared using the same protocol to that of tumor tissues[52]. T$_{CM}$ and T$_{EM}$ cells were CD3$^+$CD8$^+$CD62L$^+$CD44$^+$ and CD3$^+$CD8$^+$CD62L$^-$CD44$^+$, respectively. All these antibodies used in our experiments were diluted by ~ 200 times.

**Statistical analysis**. All the analysis data are given as mean ± SEM. The results were analyzed by the Student's $t$-test between two groups. One-way analysis of variance was used for multiple-group analysis. *$P < 0.05$ was considered significant. **$P < 0.01$ and ***$P < 0.001$ were highly significant compared with corresponding control.

**Reporting summary**. Further information on research design is available in the Nature Research Reporting Summary linked to this article.

## Data availability

All relevant data are available within the Article, Supplementary Information, Source Data file or available from the authors upon reasonable request.

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

## Acknowledgements

This work was funded by the National Basic Research Plan of China (2018YFA0208903), National Natural Science Foundation of China (81972894 and 81673364), the Ministry of Science and Technology of the People's Republic of China (2017ZX09101001006), the Fundamental Research Funds for the Central Universities (2632018ZD13), the Six Talents Summit Program of Jiangsu Province, and the Priority Academic Program Development of Jiangsu Higher Education Institutions. We thank Professor Chenhui Wang and Professor Gang Logan Liu for helpful discussion. We also thank the public platform of State Key Laboratory of Natural Medicines (China Pharmaceutical University) for assistance with cell-associated experiments.

## Author contributions

L.H. and Y.L. contributed equally to this work. C.S., L.L., and J.T. conceived the project. L.H., Y.L., and Y.Z. performed the experiments and analyzed the results. X.W., Y. Du, X. Y., and F.M. provided useful suggestions to this work. L.H., Y.L., C.S., and L.L. wrote the manuscript. C.S., L.L., and J.T. critically discussed the results and reviewed the manuscript. In Figs. 1, 4, 6, and 8, the syringe, infrared emitter (torch), mouse, and hand illustrations were drawn using Chemdraw and Microsoft Powerpoint softwares by L.H., and all other elements were produced using 3D Max software by Y. Ding. Figure 1 was finally created by L.H.

## Competing interests

The authors declare no competing interests.
