## [Peer Review File · Nature Communications]

Reviewers' comments:

Reviewer #1 (Remarks to the Author): Expert in immunology

The major finding of this paper is that a "mild-photothermal-assisted immunotherapy" has been developed by using a thermally reversible lipid gel depot which is responsive to a near-infrared photosensitizer IR820. It has also been combined with an anti-PD-L1 antibody. The anti-PD-L1 was found to be released from the gel in response to manually controlled NIR irradiation.

While this is an interesting and novel combination therapy, there are major questions which are not addressed and thus the work remains largely observational.

As one important example, the precise role of temperature is not addressed. While the authors describe a gel with a phase transition temperature in the range of 29-67 °C, and indicate that they "chose an SPC/GDO ratio of 35/65 to set the phase transition temperature at 39.5 °C" they most often target 45 °C as a "mild" thermal exposure, this is a lethal heat treatment for most cells. In any case, the precise role of temperature is not tested here. There is no thermal measurement in Fig 4, or other tumor experiments in vivo.

What temperature is actually needed for this therapy?

There are also questions about the isolation of immune cells from tumors, for example the data shown in Fig 5. The treatment groups in Fig 4 show expected variability in response. Which of these curves, and at what time point, was used for isolation of the immune cells? Did initial tumor size affect the frequency of immune cells?

The differences in CD8 +T cells in the spleen and tumor (Fig 6): There is no apparent statistical difference between the group (anti-PD-L1/1@LG+L and the group anti-PD-L1/1+L).

Reviewer #2 (Remarks to the Author): Expert in nanoparticles

This work entitled "Symbiotic Mild-photothermal-assisted in situ Immunotherapy against "Cold" Tumor via a Combined "All-in-one" and "All-in-control" Strategy" by Liping Huang describes an innovative strategy for boosting cancer immunotherapy using the effects of mild photothermal therapy in an "all-in-one" approach. It is a well-written and informed study of the complex interplay between the immune system and the tumor microenvironment, however, the primary claim of photothermal therapy was not validated in contrast to the more commonly dye-associated photodynamic therapy. As such, it requires major revisions. Below are the specific points that require attention:

1. The main use of the IR-820 dye in this study is as a photothermal therapy, however, it has also been reported to have potential as a photodynamic therapy. Indeed, many indocyanine dyes have been studied for their photodynamic therapy potential, specifically for their ability to generate reactive oxygen species. The immune-sensitizing effects reported for photodynamic therapy are highly overlapping with those reported in this work to be the result of photothermal therapy, therefore the delineation between them should be explored further. As this is the primary claim, more evidence should be included demonstrating that this is indeed a photothermal effect and not a photodynamic effect. This can be done in several ways, including, but not limited to:
 - a. Perform a similar study with molecules that have only a photothermal effect and only a photodynamic effect
 - b. Perform extensive characterization of the ratio between reactive oxygen generation and heat generation to be able to assess which may be the dominant component, perhaps they are both contributing.
 - c. Rephrase the claims of the paper, so that both possible mechanisms are accounted for.

2. This work demonstrates a masterful understanding of several complex topics, in that, it combines treatment strategies that are promisingly compatible and practical. The choice of using the near IR indocyanine dye, IR-820, however, is unclear. Specifically, in the context of the comments made above. Because it causes both radical generation and heat generation, it is not the ideal molecule to use for claiming only a photothermal response. Especially given the breadth of other molecules preferred for light-based therapies, including a host of widely used porphyrins. A brief discussion of why this molecule was chosen over the others would be highly beneficial to this work.

3. In figure 6, the efficacy of the SMPAI strategy on B16F10 tumors was evaluated, however, the survival curve was not included in the figure. This information would be pertinent to the current claims and should be included, as it is in figure 4.

4. Upregulating PD-L1 on tumors via mild PTT and suggesting this upregulation can make a "cold" tumor "hot" and primed for ICB is questionable especially given the inconsistency of tumor PD-L1 expression as a predictive biomarker for successful ICB response. The lack of a strong response at the distal tumor site is perhaps suggestive that T cells/adaptive immune system may not be as involved as proposed – rather the innate immune system may take advantage of the local antibody depot tagging tumor cells. This should be addressed/considered.

5. In the discussion, the authors suggest restrained growth at distal tumors was due to an established systemic immunity. This claim should be substantiated, potentially via re-inoculation, or addressed by another method.

Responses to reviewer 1

1. As one important example, the precise role of temperature is not addressed. While the authors describe a gel with a phase transition temperature in the range of 29-67 °C, and indicate that they "chose an SPC/GDO ratio of 35/65 to set the phase transition temperature at 39.5 °C" they most often target 45 °C as a "mild" thermal exposure, this is a lethal heat treatment for most cells. In any case, the precise role of temperature is not tested here. There is no thermal measurement in Fig 4, or other tumor experiments *in vivo*. What temperature is actually needed for this therapy?

Response: We really appreciate the reviewer for this critique. Regarding the *in vivo* temperature, we agree that 45 °C is a harmful temperature towards most cells as it is even higher than the hyperpyretic temperature, *i.e.* 42 °C. We still called it “mild photothermal therapy (PTT)” as the temperature we set was lower than what was usually used in most photothermal studies, *i.e.* 50 °C or higher. However, to judge whether an abnormally high temperature is harmful to cells, the action time should also be taken into account. In the present study, we only treated cells with a short period of 10 min, which could not cause cell ablation but was able to induce slight cytotoxicity¹. Some other literatures also reported that mild PTT with a relatively low temperature at ~45 °C was applied in tumor treatment as an aid rather than directly killing tumor cells²⁻⁴.

To verify our hypothesis, we have also carried out an experiment to test the role of the mild temperature in our study. Briefly, after seeding and adherence, cells were heated in a water bath at various temperatures, *i.e.* 37 °C, 40 °C, 43 °C and 45°C, for 10 min. Thereafter, MTT study was performed immediately or after incubation for another 24 h at an atmosphere of 37 °C and 5% CO₂. As shown in **Fig. R1 (Fig. 3a** in the revised manuscript), the cell viability was not affected in different treatments, although a small increase of cell death could be found as the temperature elevated, confirming an acceptable safety of 45 °C to treat cells for 10 min.

Additionally, western blot analysis was applied after 24 h of incubation post heat treatments. **Fig. R2 (Fig. 3b** in the revised manuscript) showed that mild heating in the range of 37-45 °C could successively upregulate the expression of PD-L1 on the surface of the tumor cells. These results definitely confirmed the rationality of our combination

strategy with aPD-L1, which was responsible for sensitizing the tumor cells to ICB treatment and realizing immune normalization in the tumor microenvironment (TME).

Conclusively, we chose heating at 45 °C for 10 min as our experimental condition of mild PTT. In *in vivo* studies, we monitored the temperature in both tumor mouse models, which were shown in **Fig. 3d-e** (for 4T1 model), and **Supplementary Fig. 10-11** (for B16F10 model).

Fig. R1 Percentage cell death of 4T1 tumor cells at 0 h or at 24 h after heating at various temperatures, *i.e.* 37 °C, 40 °C, 43 °C and 45°C, for 10 min.

Fig. R2 PD-L1 expression on 4T1 cells at 24 h after heating at various temperatures, *i.e.* 37 °C, 40 °C, 43 °C, and 45°C, for 10 min.

2. There are also questions about the isolation of immune cells from tumors, for example the data shown in Fig. 5. The treatment groups in Fig. 4 show expected variability in response. Which of these curves, and at what time point, was used for isolation of the immune cells? Did initial tumor size affect the frequency of immune cells?

Response: We apologize for not stating clearly in the previous version of this manuscript. We have conducted three parallel studies on mice bearing primary 4T1 tumors. One study (n = 5) was for tumor growth evaluation, and the results were shown in **Fig. R3b (Fig. 4b** in the revised manuscript). Another study (n = 7) was for survival rate evaluation, and the

results were shown in **Fig. R3e** (**Fig. 4e** in the revised manuscript). The third study ($n = 3$) was for immune response analysis by sacrificing the mice on day 8 after treatment, and the results were shown in **Fig. R4** (**Fig. 5** in the revised manuscript). We showed all quantified data as scattering-bar plots, which could clearly demonstrate the difference of therapeutic effects in different groups.

Fig. R3 *In situ* gel depot for inhibition of 4T1 carcinoma tumor growth *in vivo*. **a**) Schematic illustration of the animal experimental design. **b**) Primary tumor growth curve with the mean tumor volumes of 4T1 tumor-bearing BALB/c mice model. Data were

presented as mean \pm s.e.m. (n = 5). **c)** Primary tumor growth curve of individual mouse in different groups of 4T1 tumor-bearing BALB/c mice model. **d)** The tumor image obtained from the tumor-bearing BALB/c mice on day 24 after treatment. **e)** The survival percentages of the tumor-bearing BALB/c mice (n = 7).

Fig. R4 Anticancer immune response change with SMPAI strategy. a, b) DC maturation induced by SMPAI strategy on 4T1 tumor-bearing mice (gated on CD11⁺ DC cells). Cells in the lymph nodes were collected on day 8 after various treatments for assessment by flow cytometry after staining with CD11⁺, CD80 and CD86. Data

represented mean \pm s.e.m. (n = 3). **e, f**) The frequency of Tregs in tumors after different treatments examined on day 8 after treatment. **g, h**) The frequency of MDSC in tumors and spleens after different treatments examined on day 8 after treatment. **i, j**) Content of the TNF- α and IFN- γ in plasma on day 8 after treatment. The comparison of two groups was followed by unpaired student's t-test (two-tailed). *p < 0.05, **p < 0.01, ***p < 0.001.

To evaluate whether the initial tumor size affects the tumor infiltration of immunocytes, we measured the immunocyte frequencies in mice with two initial tumor sizes, *i.e.* 50 mm³ (small) and 110 mm³ (large). As a result, there was no significant difference in the proportion of immune cells (*i.e.*, CD8⁺, CD4⁺ T cells and Tregs) in the tumor tissues of both groups (**Fig. R5a**) on day 8 post the start of the treatment. In addition, the relative tumor inhibition rates of both primary tumors (treated) and distal tumors (untreated) appeared the same in both groups (**Fig. R5b-c**). We have also implanted tumors with three different sizes (small, medium, and large) on mice (n = 3). As shown in **Fig. R5d-i**, the relative proportions of CD8⁺, CD4⁺ T cells and Tregs in the tumors with different sizes only showed very small differences for all tested mice. These data declared that the initial tumor size has negligible effect on the tumor infiltration of immune cells.

Fig. R5 a) The proportions of tumor infiltrating CD8⁺, CD4⁺ T cells and Tregs. **b)** The tumor inhibition curves on tumor-bearing mice with different initial tumor sizes, *i.e.* 50 mm³ (small) and 110 mm³ (large). **c)** The relative change percentages of tumor volume calculated based on Fig. R5b. **d-f)** The photographs of different initial tumor sizes. From left to right in each photograph: Large, Medium, Small. **g-i)** the relative proportions of tumor infiltrating CD8⁺, CD4⁺ T cells and Tregs in tumors with corresponding initial sizes shown in **d, e, f**, respectively.

3. The differences in CD8⁺ T cells in the spleen and tumor (Fig 6): There is no apparent statistical difference between the group (anti-PD-L1/1@LG+L and the group anti-PD-L1/1+L).

Response: We thank the reviewer for this comment. Actually, we did find significances between the groups of “anti-PD-L1/1@LG+L” and any other groups in different organs. To increase the readability, we redrew the figures in a scattering-bar pattern and relabeled the critical significances. Please see **Fig. R6g (Fig. 8g** in the revised manuscript) for study

on B16F10 models and **Fig. R7 (Supplementary Fig. 16** in the revised manuscript) for study on 4T1 models.

Fig. R6 SMPAI strategy for the treatment of B16F10 growth *in vivo*. **a)** Fluorescence IVIS imaging describing the intratumor retention time of IR820 solution, IR820@LG with or without the NIR laser (808 nm) on B16F10 tumor-bearing mice. **b)** Quantification of the intratumor drug retention time of IR820@LG with 808 nm laser on B16F10 tumor-bearing mice. The experiment was performed once with $n = 3$ biological replicates. **c)** Schematic illustration of the animal experimental design on B16F10-tumor bearing mice. **d)** The tumor growth curve of B16F10-bearing C57BL/6 mice model. Data are shown as mean \pm s.e.m ($n = 5$). **e)** The tumor images obtained from the tumor-bearing C57BL/6 mice at 12th day after treatment. **f)** The survival percentages of the B16F10-bearing C57BL/6 mice. **g, h)** The proportions and flow cytometry plots of CD8⁺ T cells in the spleens examined on day 8 after treatment. **i, j)** The proportions and flow cytometry plots of CD8⁺ T cells in the tumors examined on day 8 after treatment. Data are shown as mean \pm s.e.m. ($n = 4$). The comparison of two groups was followed by unpaired student's t-test (two-tailed). * $p < 0.05$, ** $p < 0.01$, *** $p < 0.001$.

Fig. R7 Flow cytometric examination of the infiltration of CD4⁺ and CD8⁺ T cells in the spleens (gated on CD3⁺ T cells) of 4T1-bearing mice. Data represented mean \pm s.e.m. ($n = 3$).

Responses to reviewer 2

1. The main use of the IR-820 dye in this study is as a photothermal therapy, however, it has also been reported to have potential as a photodynamic therapy. Indeed, many indocyanine dyes have been studied for their photodynamic therapy potential, specifically for their ability to generate reactive oxygen species. The immune-sensitizing effects reported for photodynamic therapy are highly overlapping with those reported in this work to be the result of photothermal therapy, therefore the delineation between them should be explored further. As this is the primary claim, more evidence should be included demonstrating that this is indeed a photothermal effect and not a photodynamic effect. This can be done in several ways, including, but not limited to:
 - a. Perform a similar study with molecules that have only a photothermal effect and only a photodynamic effect
 - b. Perform extensive characterization of the ratio between reactive oxygen generation and heat generation to be able to assess which may be the dominant component, perhaps they are both contributing.
 - c. Rephrase the claims of the paper, so that both possible mechanisms are accounted for.
2. This work demonstrates a masterful understanding of several complex topics, in that, it combines treatment strategies that are promisingly compatible and practical. The choice of using the near IR indocyanine dye, IR-820, however, is unclear. Specifically, in the context of the comments made above. Because it causes both radical generation and heat generation, it is not the ideal molecule to use for claiming only a photothermal response. Especially given the breadth of other molecules preferred for light-based therapies, including a host of widely used porphyrins. A brief discussion of why this molecule was chosen over the others would be highly beneficial to this work.

Response to questions 1 & 2: We highly appreciate the reviewer for the kind encouragement and meanwhile providing us so many useful suggestions and critiques. We did have an in-depth discussion in our groups and completely agree with your viewpoints. To address your concerns, we designed and performed several experiments. Data and interpretation are presented below.

To investigate the potential effect of photodynamic responses on the therapeutic outcomes, we have tried to compare the reactive oxygen species (ROS) production in IR 820 aqueous solution and IR820@LG using various ROS probes, *i.e.* 9,10-anthracenediyl-bis(methylene) dimalonic acid (ABDA), dichlorodihydrofluorescein diacetate (DCFH-DA), and diphenylisobenzofuran (DPBF). As shown in **Fig. R8-10 (Supplementary Fig. 5-7)**, we interestingly found that NIR irradiation could only induce ROS production by IR820 in the solution, and negligible amount of ROS was generated by IR820 in the LG depot. Considering the extremely hypoxia environment inside the LG, this phenomenon was reasonable. Moreover, TME is usually considered hypoxia, which is also an adverse situation for ROS production. Additionally, although IR820 can be released from the LG depot, it would be easily eliminated from the tumor site during a long interval between two NIR irradiations. Thus, we can exclude IR820-induced photodynamic responses in the *in vivo* antitumor studies. These results and discussion have been added in the revised manuscript.

In current study, we need a photosensitizer to achieve mild heating. Among many potential candidates, IR820 is one of the dyes used most commonly. Even so, we also agree that IR820 may not be the best one to choose, but it performed well and did not affect the results and our conclusions in current study.

Fig. R8 The singlet oxygen production analysis with different NIR powers on IR820 solution and IR820@LG using ABDA as the singlet oxygen sensor.

Fig. R9 The singlet oxygen generation analysis by the DPBF based UV spectroscopic method (a: 0.2 W/cm², b: 0.8 W/cm²).

Fig. R10 The reactive oxygen species (ROS) generation analysis by the DCF-DA based fluorescence spectroscopic method (a: 0.2 W/cm², b: 0.8 W/cm²).

3. In figure 6, the efficacy of the SMPAI strategy on B16F10 tumors was evaluated, however, the survival curve was not included in the figure. This information would be pertinent to the current claims and should be included, as it is in figure 4.

Responses: We apologize for missing this figure in the previous version. The survival curves of mice bearing B16F10 tumors was shown below (**Fig. R11f**) and have been added in the revised manuscript (**Fig. 8f**).

Fig. R11 The survival rate curves of the B16F10-bearing C57BL/6 mice.

4. Upregulating PD-L1 on tumors *via* mild PTT and suggesting this upregulation can make a “cold” tumor “hot” and primed for ICB is questionable especially given the inconsistency of tumor PD-L1 expression as a predictive biomarker for successful ICB response. The lack of a strong response at the distal tumor site is perhaps suggestive that T cells/adaptive immune system may not be as involved as proposed – rather the innate immune system may take advantage of the local antibody depot tagging tumor cells. This should be addressed/considered.
5. In the discussion, the authors suggest restrained growth at distal tumors was due to an established systemic immunity. This claim should be substantiated, potentially via re-inoculation, or addressed by another method.

Response to questions 4 & 5: We highly appreciate the reviewer for the critiques and helpful suggestions. As described in the introduction of the revised manuscript, we hypothesized that the combination of mild PTT and anti-PD therapy would be a potential strategy to overcome the drawbacks of both PTT and immunotherapy, which may increase the immunogenicity of tumors to reprogram the “cold” TME and sensitize these tumors to immune checkpoint inhibition for synergistic anticancer therapy. We apologize for not providing substantial data to verify our hypothesis in the previous submission.

To address this issue, we have re-conducted the whole *in vivo* study on 4T1-bearing mouse models and collected more data for your reference. As shown in **Fig. R12**, the repeated *in vivo* antitumor experiment on the primary tumors and the previous one showed a highly consistent result in all groups, suggesting that the antitumor outcomes were reproducible. In addition, to clearly exhibit the differences in experimental design and

arrangement between the two *in vivo* studies, we listed a comprehensive comparison in Table 1.

Fig. R12 a) Primary tumor growth curve with the mean tumor volumes of 4T1 tumor-bearing mice model in the previous experiment. **b)** Primary tumor growth curve with the mean tumor volumes of 4T1 tumor-bearing mice model in the re-conducted experiment.

Table 1. Differences in experiment design between *in vivo* studies on 4T1-bearing mice

	Experiment or data	In previous submission	In revised manuscript
1	Primary tumor growth curves	Fig. 4B(a & b)	Fig. 4b & c
2	Primary tumor images	Fig. 4B(c)	Fig. 4d
3	Survival rates	Fig. 4B(d)	Fig. 4e
4	Body weight changes	Supplementary Fig. 11	Supplementary Fig. 14
5	Histopathological images	Supplementary Fig. 12	Supplementary Fig. 15
6	DC maturation analysis	/	Fig. 5a & b
7	T cells in primary tumors	Fig. 5A & B	Fig. 5c & d
8	T cells in spleens	Supplementary Fig. 13	Supplementary Fig. 16
9	Tregs in primary tumors	Fig. 5C Supplementary Fig. 14	Fig. 5e & f
10	MDSC in primary tumors	/	Fig. 5g Supplementary Fig. 17
11	MDSC in spleens	Fig. 5D & E	Fig. 5h
12	IL-6 in plasma	Fig. 5F	Supplementary Fig. 18
13	TNF- α in plasma	Fig. 5G	Fig. 5i
14	IFN- γ in plasma	Fig. 5H	Fig. 5j
15	Distal tumor growth curves	Fig. 4C Supplementary Fig. 10	Fig. 6b & c
16	T cells in distal tumors	/	Fig. 6d & f
17	Tregs in distal tumors	/	Fig. 6e & g
18	CD8⁺/Treg ratios	/	Fig. 6h
19	Lung metastasis w/o tumor	Fig. 4D	/

	cell infusion		
20	Lung metastasis w/ tumor cell infusion	/	Fig. 7b
21	H&E staining of lung tissues	/	Fig. 7c
22	Metastasis nodules in lungs	/	Fig. 7d
23	T _{EM} in spleens	/	Fig. 7e
24	T _{EM} in the lymph nodes	/	Fig. 7f

Firstly, we investigated whether the SMPAI strategy could produce tumor antigens and promote DC maturation in the inguinal lymph nodes. Matured DCs can present the major histocompatibility complex-peptide to T-cell receptor when arriving at lymph nodes, which plays a crucial role in initiating and regulating the adaptive immunities⁵. It was found that the percentage of matured DCs (CD11c⁺CD80⁺CD86⁺) in the “aPD-L1/I@LG+L” group, *i.e.* ~42%, was significantly higher than that in the “PBS” group, *i.e.* ~23%. As a comparison, the DC maturation percentages were ~38%, ~30% and ~33% in the “IR820@LG+L”, “aPD-L1/I@LG-L” and “aPD-L1/I+L” groups, respectively (**Fig. R13** shown below and **Fig. 5a,b** in the revised manuscript). These results illustrated that the SMPAI strategy was capable of inducing a strong immune stimulation effect.

Fig. R13 DC maturation induced by the SMPAI strategy on 4T1 tumor-bearing mice (gated on CD11⁺ DC cells). Cells in the lymph nodes were collected on day 8 after various treatments for assessment using flow cytometry after staining with CD11c⁺, CD80 and CD86.

Secondly, the distal tumor model was established to evaluate if the active immune responses were strong enough to inhibit the untreated distal tumors (**Fig. R14a** shown below and **Fig. 6a** in the revised manuscript). After local treatment, the size change of the distal tumors was recorded and plotted (**Fig. R14b, c** and **Fig. 6 b, c** in the revised manuscript). The results showed that all distal tumors grew at a similar rate in the first

several days. Strikingly, the growth of the distal tumors on the mice treated with formulations containing aPD-L1, *i.e.*, the “aPD-L1/I@LG-L”, “aPD-L1/I+L”, and “aPD-L1/I@LG+L” groups, was significantly slower than that of the groups without aPD-L1 after two weeks, which confirmed that aPD-L1 played a key role in the systemic immunity establishment as well as the distal tumor inhibition. However, we found that the abscopal immune outcomes of single treatments, *i.e.*, the “IR820@LG+L” and “aPD-L1/I@LG-L” groups, was limited on the treatment of distal tumors, suggesting that systemic immunity was likely disabled in the immunosuppressive TME of the distal tumors without a combined therapy. In addition, it was found that the “aPD-L1/I@LG+L” group achieved much better efficacies in distal tumors than the “aPD-L1/I+L” group, indicating that free drugs without LG depot could not maintain a long-term immune treatment effect. Based on these results, we further confirmed that the long-term abscopal immune responses to rebuild the “cold” TME is of significant importance to realize ideal ICB outcomes.

To better demonstrate the improved tumor inhibition effect of the aPD-L1 and mild PTT combination on distal tumors, the intratumor infiltration of CD8⁺ CTLs and Tregs were examined by flow cytometry. The frequency of tumor-infiltrating CD8⁺ CTLs in distal tumors of the “aPD-L1/I@LG+L” group was 4.7 folds higher than that of the “PBS” group, and 1.6 folds higher than that of the “aPD-L1/I+L” group (**Fig. R14d-f** shown below and **Fig. 6d-f** in the revised manuscript). Meanwhile, Treg frequency in the “aPD-L1/I@LG+L” group was much lower than that in the “PBS” group, verifying that the SMPAI strategy significantly reversed the immune tolerance of distal tumors (**Fig. R14e-g** shown below and **Fig. 6e-g** in the revised manuscript). Importantly, the CD8⁺/Treg ratio were greatly enhanced in distal tumors of the mice treated with aPD-L1/I@LG+L when compared to the other groups (**Fig. R14h** shown below and **Fig. 6h** in the revised manuscript), which were more important to evaluate the antitumor immune responses.

Fig. R14 The therapeutic effect on abscopal tumors with SMPAI strategy. **a)** Schematic illustration of the animal experimental design for distal tumors. **b)** Distal tumor growth curve with the mean tumor volumes of 4T1 tumor-bearing BALB/c mice model. **c)**

Distal tumor growth curve of individual mouse in different groups of 4T1 tumor-bearing BALB/c mice model. **d)** Representative flow cytometry plots showing different groups of T cells in distal tumors (gated on CD3⁺ T cells). **e)** Representative flow cytometry plots showing percentages (gated on CD4⁺ cells) of Tregs in distal tumors on day 8 after different treatments. **f)** Proportions of the intratumor infiltration of CD8⁺ killer T cells in distal tumors. **g)** The frequency of Tregs in distal tumors upon various treatments. **h)** CD8⁺ CTL: Treg ratios in the distal tumors after various treatments. Data are represented as the mean \pm s.e.m. (n = 3).

Thirdly, we further evaluated the therapeutic efficacy of our strategy on treating a more aggressive whole-body spreading tumor model (**Fig. R15** shown below and **Fig. 7** in the revised manuscript). In this study, mice were intravenously injected with 4T1 cells on day 30 after the start of various treatments (**Fig. R15a** shown below and **Fig. 7a** in the revised manuscript). After feeding the mice for another 21 days, the lung tissues in different groups were harvested for metastasis analysis. As shown in **Fig. R15b,c**, metastatic foci in lungs was observed in all mice except in the “aPD-L1/I@LG+L” group, suggesting the success of our SMPAI strategy in inhibiting the lung metastasis. In addition, significantly more pulmonary metastasis nodules were observed in all control groups than in the “aPD-L1/I@LG+L” group (**Fig. R15d** shown below and **Fig. 7d** in the revised manuscript). These results further declared that the LG-based “all-in-control” strategy was beneficial for the establishment of active immune responses to the metastatic 4T1 tumors.

Finally, we tried to analyze the adaptive immunity establishment at cellular levels. As is well known, memory T cells are classified as central memory T cells (T_{CM} cells: CD11b⁺CD8⁺CD44⁺CD62L⁺) and effector memory T cells (T_{EM} cells: CD11b⁺CD8⁺CD44⁺CD62L⁻). While T_{CM} cells only provide protections after antigen-stimulated clonal expansion, differentiation, and trafficking, T_{EM} cells can induce immediate protections by producing cytokines, such as IFN- γ ⁵⁻⁷. Therefore, to understand the mechanism underlying the tumor-specific anti-metastasis property of the SMPAI strategy, the T_{EM} in the spleen and lymph nodes were examined using flow cytometric measurement on the next day after intravenous infusion of 4T1 cells. The mice treated with aPD-L1/I@LG+L showed significantly increased T_{EM} frequency both in the spleens and in lymph nodes, whereas the “PBS”, “aPD-L1@LG+L”, “aPD-L1/I@LG-L” and “aPD-

L1/I+L” groups exhibited moderate generation of T_{EM} cells (Fig. R15e,f shown below and Fig. 7e,f in the revised manuscript), suggesting that the SMPAI strategy could produce a better memory immunity effect that was helpful to inhibit cancer metastasis.

Fig. R15 Metastasis prevention via SMPAI-induced long-term immune effects. a) Schematic illustration for SMPAI-mediated inhibition of tumor metastasis. **b)** Representative photographs of lung tissue with tumor metastasis. **c)** H&E staining of the lung tissue collected on day 51 (Scale bars: 100 μ m). **d)** Quantification of pulmonary

metastasis nodules in different groups of 4T1 tumor-bearing BALB/c mice. e,f) Proportions of effector memory T cells (T_{EM}) in the spleens and lymph nodes (gated on CD3⁺ CD8⁺ T cells) examined after 1 day post intravenous infusion of the 4T1 cells. Data are presented as mean \pm s.e.m. (n = 3). The comparison of two groups was followed by unpaired student's t-test (two-tailed). *p < 0.05 and **p < 0.01.

References

1. Ware MJ, *et al.* A new mild hyperthermia device to treat vascular involvement in cancer surgery. *Scientific Reports* **7**, 11299 (2017).
2. Zhang X, *et al.* Efficient Near Infrared Light Triggered Nitric Oxide Release Nanocomposites for Sensitizing Mild Photothermal Therapy. *Advanced Science* **6**, 1801122 (2019).
3. Peng J, *et al.* Photosensitizer Micelles Together with IDO Inhibitor Enhance Cancer Photothermal Therapy and Immunotherapy. *Advanced Science* **5**, 1700891 (2018).
4. Yang Y, *et al.* 1D Coordination Polymer Nanofibers for Low-Temperature Photothermal Therapy. *Advanced materials* **29**, (2017).
5. Chen Q, Xu L, Liang C, Wang C, Peng R, Liu Z. Photothermal therapy with immune-adjuvant nanoparticles together with checkpoint blockade for effective cancer immunotherapy. *Nature Communications* **7**, 13193 (2016).
6. Wherry EJ, *et al.* Lineage relationship and protective immunity of memory CD8 T cell subsets. *Nature Immunology* **4**, 225-234 (2003).
7. Kaech SM, E John W, Raft A. Effector and memory T-cell differentiation: implications for vaccine development. *Nature Reviews Immunology* **2**, 251-262 (2002).

REVIEWERS' COMMENTS:

Reviewer #1 (Remarks to the Author):

I believe the authors have done a good job of revising their paper with additional experimentation which address most of my concerns.

However, I feel strongly that the authors should make a comment in the Discussion about the need for more complete temperature-effect relationships in vivo in future studies.

Reviewer #2 (Remarks to the Author):

The authors have addressed all of the outstanding issues.

Reviewer #1: I believe the authors have done a good job of revising their paper with additional experimentation which address most of my concerns. However, I feel strongly that the authors should make a comment in the Discussion about the need for more complete temperature-effect relationships in vivo in future studies. Response: Thank the reviewer for the very helpful suggestion. We have added a comment in the Discussion about the need for more complete temperature-effect relationships in vivo in future studies. We revised the related part as “In the current study, the precise control of mild PTT temperature is not only important to adjust the release of drugs, but is also critical in sensitizing the immunodeficient tumors as well as potentiating the anti-PD-L1 treatment. Therefore, in future studies, it is essential to establish more complete temperature-effect relationships in vivo for optimal therapeutic efficacy.”, which was highlighted in red in the second paragraph in the Discussion